TOOLS

# Surface Morphometrics reveals local membrane thickness variation in organellar subcompartments

Michaela Medina[1]*, Ya-Ting Chang[1]*, Hamidreza Rahmani[1], Mark Frank[2], Zidan Khan[3], Daniel Fuentes[1], Frederick A. Heberle[3], M. Neal Waxham[4], Benjamin A. Barad[1,2], and Danielle A. Grotjahn[1]

Lipid bilayers form the basis of organellar architecture, structure, and compartmentalization in the cell. Decades of biophysical, biochemical, and imaging studies on purified or *in vitro*–reconstituted liposomes have shown that variations in lipid composition influence the physical properties of membranes, such as thickness and curvature. However, similar studies characterizing these membrane properties within the native cellular context have remained technically challenging. Recent advancements in cellular cryo-electron tomography (cryo-ET) imaging enable high-resolution, three-dimensional views of native organellar membrane architecture preserved in near-native conditions. We previously developed a "Surface Morphometrics" pipeline that generates surface mesh reconstructions to model and quantify cellular membrane ultrastructure from cryo-ET data. Here, we expand this pipeline to measure the distance between the phospholipid head groups of the membrane bilayer as a readout of membrane thickness. Using this approach, we demonstrate thickness variations both within and between distinct organellar membranes. We show that organellar membrane thickness positively correlates with other features, such as membrane curvedness, in cells. Further, we show that subcompartments of the mitochondrial inner membrane exhibit varying membrane thicknesses that are independent of whether the mitochondria are in fragmented or elongated networks. We also demonstrate that our technique, when applied to three-dimensional data, yields results that match existing measurements obtained from two-dimensional data of *in vitro* samples. Finally, we demonstrate that large membrane-associated macromolecular complexes exhibit distinct density profiles that correlate with local variations in membrane thickness. Overall, our updated Surface Morphometrics pipeline provides a framework for investigating how changes in membrane composition in various cellular and disease contexts affect organelle ultrastructure and function.

## Introduction

Eukaryotic cells rapidly remodel their organellar membranes in response to a variety of cellular stresses and physiological conditions. Lipid bilayers, the core unit of organellar membranes, are created through the self-assembly of amphipathic phospholipids that orient their hydrophilic head groups to shield their hydrophobic tails from the aqueous cytoplasmic environment. Their dynamic and fluid nature enables membrane remodeling required for many diverse cellular processes, including inter-organellar communication, scaffolding, organelle biogenesis, and quality control. In addition to phospholipids, integral membrane proteins also dictate the inherent properties of organellar membranes. Membrane proteins such as ATP synthase dimers have the capacity to deform, bend, and stabilize the phospholipid bilayer (Blum et al., 2019). Conversely, membrane properties such as stiffness and curvature can impact the folding, localization, and function of integral membrane proteins (Stamou et al., 2015). Through dynamic remodeling of lipid and

membrane protein composition, organelles can modulate their shape and thus function to facilitate cellular processes, such as vesicular trafficking (Bonifacino and Glick, 2004; Watson and Stephens, 2005), inter-organellar resource transfer (Sassano et al., 2023), and organellar fission and fusion (Westermann, 2010; Youle and van der Bliek, 2012). Despite this important link between membrane remodeling and adaptive function, defining local changes to lipids and proteins in their native context has remained technically challenging.

Under conditions of low dose and defocus (Heberle et al., 2023), cryo-electron microscopy (cryo-EM) imaging has the resolving power to distinguish the two opposing rows of phospholipid head groups (PHG) that comprise the lipid bilayer and the proteins embedded within them. Several groups have harnessed the power of cryo-EM to reveal how different compositions of phospholipids, when assembled in unilamellar vesicles, can impart distinct membrane properties, including membrane

....................................................................................................................................................................................................................

[1]Department of Integrative Structural and Computational Biology, The Scripps Research Institute, La Jolla, CA, USA;   [2]Department of Chemical Physiology and Biochemistry, School of Medicine, Oregon Health & Science University, Portland, OR, USA;   [3]Department of Chemistry, University of Tennessee, Knoxville, TN, USA;   [4]Department of Neurobiology and Anatomy, McGovern Medical School, UTHSC-Houston, Houston, TX, USA.

*M. Medina and Y.-T. Chang contributed equally to this paper.   Correspondence to Danielle A. Grotjahn: grotjahn@scripps.edu;   Benjamin A. Barad: barad@ohsu.edu.

thickness, rigidity, and compressibility (Heberle et al., 2020; Heberle et al., 2023; Sharma et al., 2023). In combination with cell thinning techniques such as cryo-focused ion beam (FIB) milling, it is now possible to obtain high-resolution views of lipid bilayers within their native cellular context using cryo-electron tomography (cryo-ET) imaging. There are also multiple voxel-based density sampling strategies developed to analyze protein structure and reveal the organization of the embedded proteins within membranes visible in cellular cryo-ET data (Martinez-Sanchez et al., 2020; Lamm et al., 2022). However, to date, none of these density sampling approaches have been adapted to calculate membrane thickness directly and specifically. A recently reported voxel segmentation-based approach (Glushkova et al., 2025, *Preprint*) estimates membrane thickness by generating and converting voxel segmentations of membranes into oriented point clouds to approximate the edge points as the membrane "boundaries." Ray projections between opposing boundary edge points are then used as a proxy to estimate membrane thickness. While this approach revealed relative changes in thickness, it led to measurements considerably larger than previously reported for *in vitro* vesicles. An approach that allows correlation with other features of membrane geometry would be beneficial.

Here, we present a new method for measuring membrane thickness using triangulated surface mesh reconstructions to calculate voxel density–based line scans across organellar membranes. Surface mesh reconstructions provide a more accurate model of the inherent geometry of the membrane, independent of voxel size. By calculating per-triangle density line scans across these surface mesh models, we generate plots of local membrane density extracted from the tomogram itself. We demonstrate that the opposing PHG of the membrane bilayer result in peaks within these line scans; we use the distance between these peaks as the membrane thickness, consistent with previous studies (Heberle et al., 2020). This model-guided approach also integrates with other ultrastructural measurements in the Surface Morphometrics pipeline, enabling correlation with geometric features such as curvature and intermembrane spacing. We show that this approach can be used to detect statistically significant differences in membrane thickness across cell types, organelles, and their functionally distinct membrane compartments. Importantly, we demonstrate that our method applied to 3D tomographic data yields thickness values consistent with established measurements from 2D projection images of purified vesicles. Furthermore, we demonstrate that large membrane-associated macromolecular complexes exhibit distinct density profiles that correlate with local variations in membrane thickness. By integrating these density-based measurements within our existing Surface Morphometrics pipeline (Barad et al., 2023), we provide the first automated approach to measure intracellular membrane thickness and correlate it with the rest of the cellular context.

## Results

### Measuring organellar membrane thickness in cellular cryo-electron tomograms

We set out to develop an accurate and automated pipeline for measuring organellar membrane thickness from cellular cryo-

ET data directly (Fig. 1). Our desired measurement of membrane thickness is the distance between the two PHG—this measurement has been used previously for measurements in *in vitro* vesicles (Heberle et al., 2020; Seneviratne et al., 2023). Because many tomograms resolve membrane leaflets, we believe this is the most precise method to estimate membrane thickness *in situ*. We generated and collected tilt series of thin (80–150 nm) lamellae at relatively low defocus (4–6 μm) of vitrified mouse embryonic fibroblasts (MEFs) expressing mitochondria-targeted GFP (MEFmtGFP) in order to maximize the ability to resolve individual leaflets of membranes. We performed automated segmentation of all visible membranes (Lamm et al., 2024, *Preprint*) and manually separated the different cellular membranes, such as the outer and inner mitochondrial membranes (OMM and IMM, respectively), endoplasmic reticulum (ER) membranes, and vesicles. The resulting voxel segmentation models were processed through the Surface Morphometrics pipeline (Barad et al., 2023) to generate triangulated surface meshes that approximate the mid-surface between the two sides of the bilayer (Fig. 1, A and B).

To measure thickness, we start from the midpoint of each triangle on the surface mesh and then systematically interpolate the tomogram density along a 10-nm path in both the positive and negative directions along the normal vector (Fig. 1 B). We use these measurements to generate a tomogram density line scan, which reveals two peaks corresponding to the densities associated with the PHG (Fig. 1 C). We estimate the distance between these head groups by fitting a dual Gaussian distribution to the two peaks, with the central position of each Gaussian defined as one of the PHG. We found that estimating these peaks from non-denoised Warp back-projected tomograms is challenging and prone to artifacts due to the low signal-to-noise ratio. To enhance signal-to-noise, we performed distance-weighted averaging of the signal of triangles within a 12-nm radius (Fig. 1 D), hyperparameters which we determined had a good balance of local information with robust measurement of thickness. We have opted to make this radius user-configurable in the Surface Morphometrics pipeline in case different averaging is preferred for systems with better or worse signal-to-noise. We used this approach to measure PHG distance locally across every triangle of each surface mesh reconstruction in our dataset and show the different PHG distances between different organelles, revealing an apparent increase in thickness in the IMM compared with the OMM (Fig. 1 E).

To test the robustness of this approach to different data processing conditions, we performed this analysis on a range of binned tomograms (6.65, 9.98, and 13.30 Å/pixel) (Fig. S1, A–C). We found that using binned tomograms (9.98 Å/pixel) yielded segmentation outputs that faithfully segmented the underlying membrane density visible in the tomogram, without artifacts due to protruding membrane proteins or blurring of the two membrane leaflets (Fig. S1 C). Therefore, we concluded that, for this set of data, tomograms reconstructed to 9.98 Å/pixel provide the best balance between accuracy of segmentation and clear resolution of lipid bilayers. Upon inspection of the generated surfaces, we observed less reliable thickness measurements at the edges of

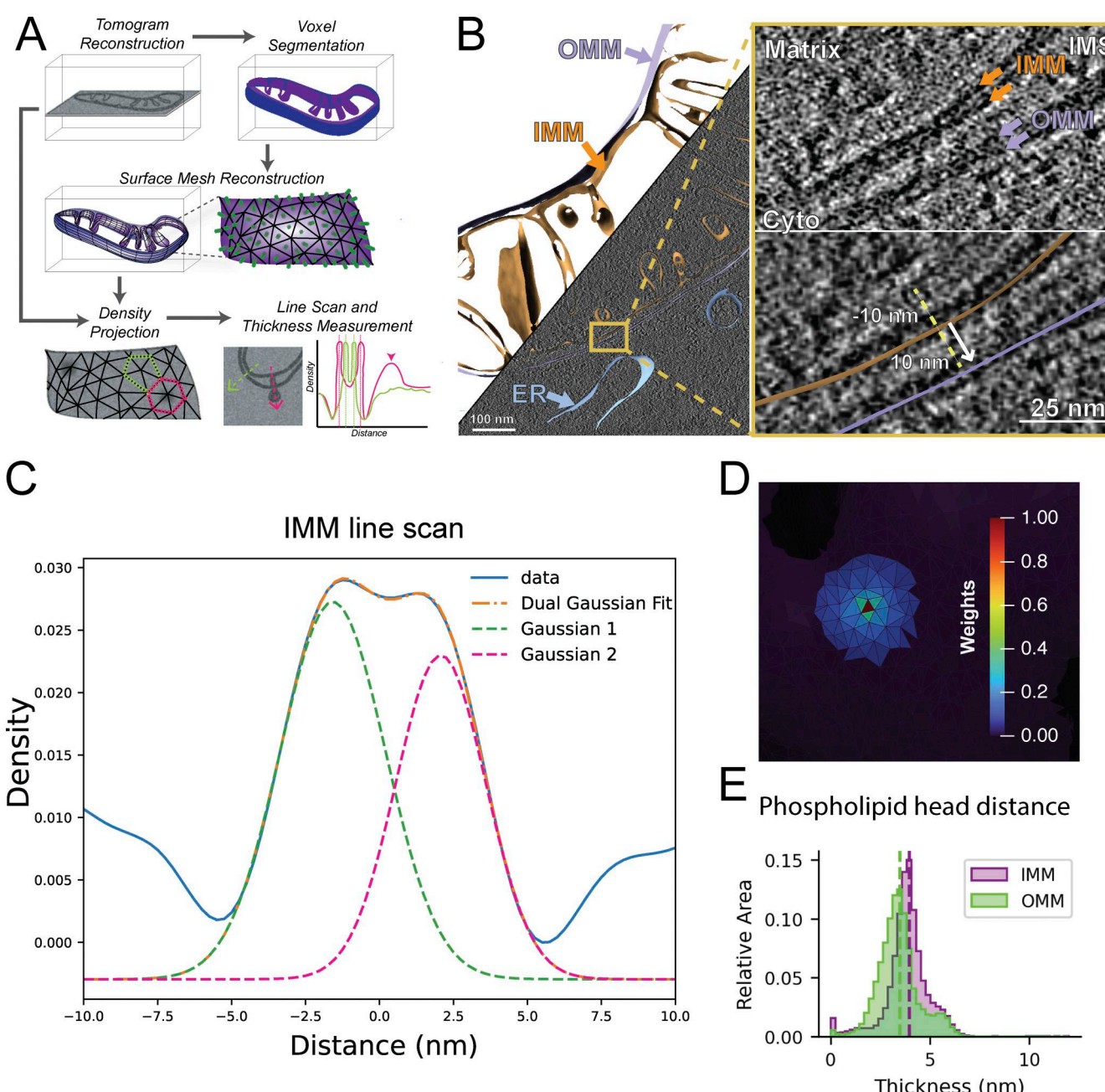

Figure 1. **Extending Surface Morphometrics to measure lipid bilayer thickness within tomograms. (A)** Tomograms were reconstructed and segmented to produce voxel segmentations that labeled distinct cellular membranes. Voxel segmentations were processed through the Surface Morphometrics pipeline (Barad et al., 2023) to generate triangulated surface meshes. The tomogram density was interpolated systematically from the midpoint of each triangle on the surface mesh, following the normal vector in both the positive and negative directions (dashed arrows). These measurements are used to generate a voxel line scan, which reveals two peaks corresponding to the densities associated with the PHG (dashed vertical lines). We calculated the distance between these peaks as a readout of membrane thickness. Line scans that are extended beyond the lipid bilayer show additional peaks representing protein macromolecules (pink arrowhead). **(B)** Triangle mesh models for both IMM (orange) and OMM (lavender) are overlaid on top of the binned tomogram reconstruction, which is of sufficient resolution to distinguish lipid head groups visually. The meshes can be used to generate local vectors for density line scan analysis within the to-mogram. **(C)** Line scans generated by sampling along the normal vectors can be combined across a surface to generate a smooth plot with peaks corresponding to the two head groups of the bilayer (blue line). We fit the curve with a dual Gaussian model (orange dashed line, individual Gaussians, pink and green) and used the peak-to-peak distance as the bilayer distance. **(D)** Local thickness variation analysis was determined via a local weighted averaging scheme, in which the line scan of the central triangle (red) is combined with neighboring triangles up to 12 nm away with weights that decrease with distance from the center. **(E)** Using per-triangle thickness measurements, we identified the differences in the distribution of thicknesses within different membranes, as depicted in a triangle area–weighted histogram, whose y axis measures the relative area of membrane in each thickness bin.

the surface meshes due to artifacts caused by the missing wedge (Fig. S1, D and E). To ensure that we only included membranes that had reliable thickness estimations, we implemented an edge exclusion feature to exclude all triangles within 8 nm of the edge of the original surface. This led to robust and accurate membrane thickness measurements that are free from artifacts inherent to cryo-ET data. All reported membrane thickness measurements were generated with edge filtering applied to the surface mesh reconstructions.

## Cellular membranes display significant differences in membrane thickness

Previous biochemical and structural analyses of purified and reconstituted membranes have demonstrated that membranes with unique lipid and protein compositions result in changes to the biophysical properties of the lipid bilayer, such as thickness (Andersen and Koeppe, 2007). To assess whether these differences are observed in the native cellular environment, we applied our pipeline to calculate membrane thickness on a per-triangle basis across all triangles within each surface mesh for every organellar membrane in our dataset. Plotting the spatial distribution of these thicknesses directly on the surfaces (Fig. 2, A and B; and Fig S2) and the combined distribution of thicknesses for all surface mesh triangles on a histogram (Fig. 2 C) revealed subtle differences in thickness across cellular organelles. Assessing the median thicknesses as individual observations for each organelle showed significant differences in membrane thickness across cellular compartments (Fig. 2 D). Interestingly, our results show that the OMM is significantly thinner (3.2 ± 0.10 nm) compared with the IMM (3.6 ± 0.07 nm), demonstrating that variations in membrane thickness exist even within the same organelle. The OMM also showed statistically significant reductions in membrane thickness relative to ER (3.7 ± 0.05 nm) membranes.

To further assess the performance of our pipeline across distinct organellar membranes and cell types, we calculated membrane thickness on tomograms of yeast (*Saccharomyces cerevisiae*) cell lamella with visible plasma membrane, vacuole membrane, and nuclear envelope bilayers from both new data and a previously published dataset (Electron Microscopy Public Image Archive [EMPIAR] 12534) (Chang et al., 2025). Consistent with our analysis in MEF cells, we detected significant differences in membrane thickness among various cellular membranes (Fig. 2, E–H). We observed a similar trend in membrane thickness variations in yeast as in MEF cells, with OMM (2.8 ± 0.27 nm) being significantly thinner than IMM (3.4 ± 0.13 nm), ER (3.8 ± 0.15 nm), and vesicles (4.1 ± 1 nm). In addition, we observe membrane thickness variations across the bilayers of the plasma membrane (4.2 ± 0.16 nm), vacuole membrane (4.1 ± 0.18 nm), and nuclear envelope (3.5 ± 0.23 nm), with the plasma membrane exhibiting the largest thickness values. Taken together, we show that different organellar membranes exhibit significant differences in average membrane thickness in multiple species, demonstrating the power of our approach to quantify subnanometer-level differences in membrane thickness across membranes visualized in their native environment.

## Subcompartments of the IMM vary in membrane thickness

We next asked whether variations in membrane thickness are observed across functionally distinct regions within the same organellar membrane. Specialized lipids and proteins influence the shape of the IMM and help fold it into distinct subcompartments, including the regions closely appressed to the OMM, termed the inner boundary membrane (IBM), the protruding regions called the crista body, and the transition zones between these regions, called the crista junctions. We previously showed that our Surface Morphometrics pipeline can automatically classify between these distinct compartments based on their distance from the OMM (Barad et al., 2023). We performed a similar subclassification procedure on the IMM in this dataset and measured the membrane thickness locally within each of these subcompartments (Fig. 3 A). Interestingly, we observed significant differences in the thickness of these compartments, with the crista body exhibiting significantly thicker membranes (3.8 ± 0.04 nm) relative to the crista junction and IBM compartments (Fig. 3, B and C). While subtle, this significant change in membrane thickness across the contiguous membrane suggests that these differences may reflect an additional structural regulation of IMM subcompartment specialization.

Like the IMM, the ER membrane can be further subdivided into two spatially and structurally distinct compartments: the rough ER, comprised of sheetlike structures studded with co-translating ribosomes, and the smooth ER, which forms tubular projections that often make functional contacts with other organelles (Wang et al., 2015; Garfield and Cardell, 1987). We leveraged these distinct structural characteristics to classify the two types of membranes in our dataset based on the presence of membrane-docked ribosomes (rough ER) or the absence of ribosomes (smooth ER) (Fig. 3 D). In contrast to the subcompartments of the IMM, we detected no significant differences between the smooth and rough ER, with membrane thicknesses of 3.7 ± 0.09 nm and 3.7 ± 0.22 nm, respectively (Fig. 3, E and F).

Given the observed IMM compartment-specific differences, we wondered whether other aspects of mitochondrial structure are associated with changes in membrane thickness. Mitochondria form large, dynamic networks that can exhibit distinct cellular distributions, including highly elongated (i.e., hyperfused) or fragmented networks. We previously demonstrated that these distinct network morphologies are associated with significantly distinct membrane ultrastructures that vary in inter- and intramembrane spacing, curvature, and orientation (Barad et al., 2023). We set out to understand whether a similar connection exists between bulk mitochondrial morphology (i.e., elongated versus fragmented) and mitochondrial membrane thickness. We performed cryo-fluorescence microscopy to classify the network morphology of each cell prior to cryo-FIB milling and cryo-ET acquisition (Fig. S3 A) (Barad et al., 2023). We calculated the average membrane thickness across mitochondrial membrane surfaces and observed no statistically significant differences in the thickness of the OMM or IMM (and IMM subcompartments) based on network morphology (Fig. S3 B), in contrast with previous work showing differences in intermembrane spacing and curvature (Barad et al., 2023).

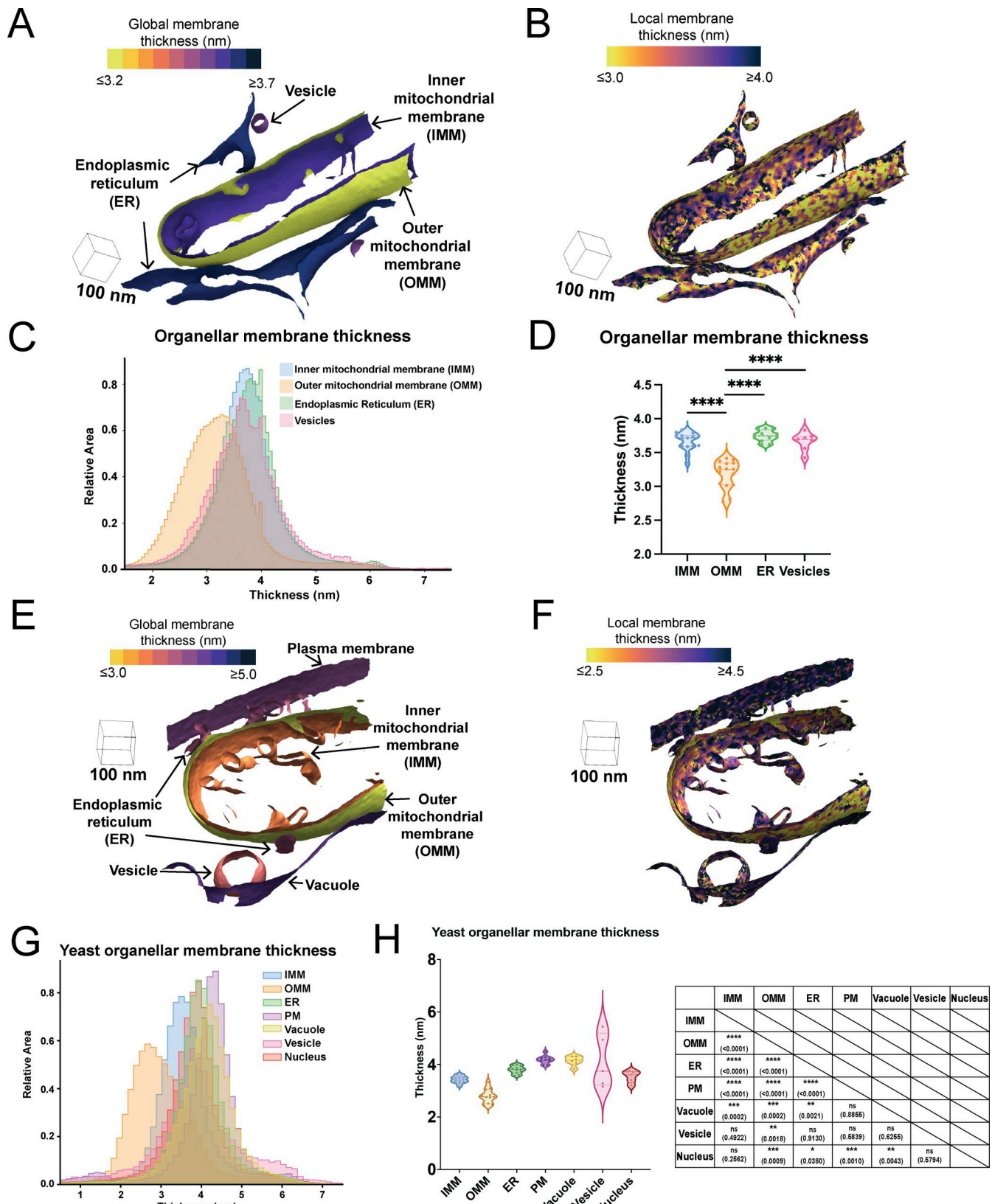

Figure 2. **Membrane thickness varies across organelles. (A)** Global per-triangle measurements across different organelle classes are mapped to the different organelle classes observed in a single tomogram from a lamella generated from a MEF cell. **(B)** Local per-triangle measurements across different organelle classes are mapped to the different organelle classes observed in a single tomogram from MEF. **(C)** Per-triangle median thickness histograms highlight the difference in thickness distributions between different physiological organelles in MEF. **(D)** Violin plot of per-surface median thickness reveals statistically significant differences in thickness between different organelles in MEF. IMM: $n = 15$; OMM: $n = 15$; ER: $n = 12$; vesicle: $n = 7$. P values from the Mann–

Whitney U test are indicated. ****P < 0.001. **(E)** Global per-triangle measurements across different organelle classes are mapped to the different organelle classes observed in a single tomogram from yeast (*S. cerevisiae*). **(F)** Local per-triangle measurements across different organelle classes are mapped to the different organelle classes observed in a single tomogram from yeast cell lamellae. **(G)** Per-triangle median thickness histograms highlight the difference in thickness distributions between different physiological organelles in yeast cell lamellae. **(H)** Violin plot of per-surface median thickness reveals statistically significant differences in thickness between different organelles in yeast. IMM: $n = 11$; OMM: $n = 11$; ER: $n = 11$; plasma membrane: $n = 9$; vacuole: $n = 6$; vesicle: $n = 5$; nucleus: $n = 5$. P values from the Mann–Whitney U test are shown in the table. PM, plasma membrane.

## Membrane thickness positively correlates with membrane curvature

Previous studies using *in vitro*–reconstituted membranes demonstrated that membrane rigidity (i.e., the resistance to curvature) increases with thickness (Bermúdez et al., 2004). We observe significant variability in the curvature of the IMM, with the crista junction and the crista "tip" exhibiting the highest degree of curvature relative to the crista body and IBM (Fig. 4 A) (Barad et al., 2023). Therefore, we wondered whether increases in membrane curvature are associated with thinner membrane regions in organellar membranes within the native cellular environment. To test this, we calculated the curvature of all triangles within the IMM surface mesh reconstructions and partitioned the curvature values into separate quantile ranges for comparison (0–0.5, 0.5–0.9, 0.9–0.95, 0.95–0.99, 0.99–1). Interestingly, plotting the membrane thickness for each triangle within the curvature quartiles showed that triangles with the highest curvature (0.99–1 quantile) were associated with significantly thicker membranes relative to medium and low curvature regions (Fig. 4 B).

Given that ATP synthase dimers have the capacity to induce and localize to regions containing high degrees of membrane curvature (Davies et al., 2012), we wondered whether the presence of ATP synthase in our datasets was associated with local changes in membrane thickness. In species like *Chlamydomonas reinhardtii*, ATP synthase is primarily organized as dimer rows enriched at the crista tip regions. In our dataset, we observe clear densities for ATP synthase that are assembled in a mixture of dimers and monomers, both in the crista tip and in the crista body regions (Fig. 4 C). To investigate the association of these ATP synthase complexes with regions of high curvature and membrane thickness, we manually selected 8,993 ATP synthase monomers from a larger dataset and performed subtomogram averaging to refine the position and orientation of these particles. This resulted in a structure (resolved to 13 Å) resembling ATP monomers from previous reports (Nesterov et al., 2020) (Fig. 4 D and Fig. S3 C). We applied a "patch-based" analysis (Chang et al., 2025) to subselect local patches of surface mesh reconstructions that correspond to the membrane "footprint" of ATP synthase on the crista body region of the IMM, where ATP synthases are predominantly localized. In brief, this involved identifying the nearest IMM surface triangles of each ATP synthase particle coordinate and extracting the surrounding triangles within a radius of 120 Å of those nearest triangles (Fig. 4 E). Calculating membrane thickness locally at each ATP synthase patch region revealed no significant differences in membrane thickness compared with patches generated at random positions within the crista body (Fig. 4 F). However, in the context of curvature, we observed that those ATP synthase patches at high-

curvature regions were also associated with thicker membranes, compared with what would be expected from random chance (Fig. 4 G). This suggests that individual ATP synthase particles associated with high-curvature regions are also found in regions of greater membrane thickness.

## Thickness measurements from the Surface Morphometrics pipeline on 3D reconstructed data closely match those from previous approaches on 2D data of an *in vitro* vesicle sample

To determine the degree to which changes might be due to artifacts from tomography data collection and to benchmark our measurements against alternative imaging modalities, we performed thickness calculations on a sample of *in vitro*–extruded vesicles from untilted (2D projections) cryo-EM micrographs using previously described approaches (Heberle et al., 2020; Heberle et al., 2023) and on tomographic data (3D reconstructions) using our Surface Morphometrics pipeline (Fig. 5). To ensure consistency across imaging and sample conditions, we collected these data from the same sample deposited on the same electron microscopy (EM) grid. Both methods revealed thickness variations between 3 and 4 nm, though more variation was measured in the tomographic data, likely due to increased defocus, artifacts from tomogram reconstruction, and reduced signal-to-noise (Fig. 5, B–E). Despite these differences in variation, the two techniques showed remarkably similar overall thickness measurements, with a median of 3.56 nm for cryo-EM and 3.52 nm for cryo-ET. This results in a difference of <2%, suggesting that the measurements made by the Surface Morphometrics pipeline are consistent with state-of-the-art techniques for measuring bilayer thickness using other imaging modalities. Because these *in vitro* vesicles contained no proteins, we used them to test the degree to which thickness and curvature correlate in the absence of protein factors (Fig. 5, G and H). We observed no statistically significant correlation between local membrane curvature and overall vesicle radius, suggesting that the increased thickness at high curvature in Fig. 4 G may be a specific feature of cellular membranes, possibly due to curvature-associated lipids or proteins.

## Local changes in membrane thickness are colocalized with membrane-associated proteins that exhibit unique density line scan profiles

Unique to our approach is the ability to accurately identify local variability in membrane thickness on a locally averaged per-triangle basis (Fig. 2 B, Fig. 6 A, and Fig. S2). Within our surfaces, we isolated the local patches that exhibited the most increased or reduced membrane thickness relative to the surrounding areas (Fig. 6 A and Fig. S4). Mapping these local regions back to their locations within the tomogram revealed

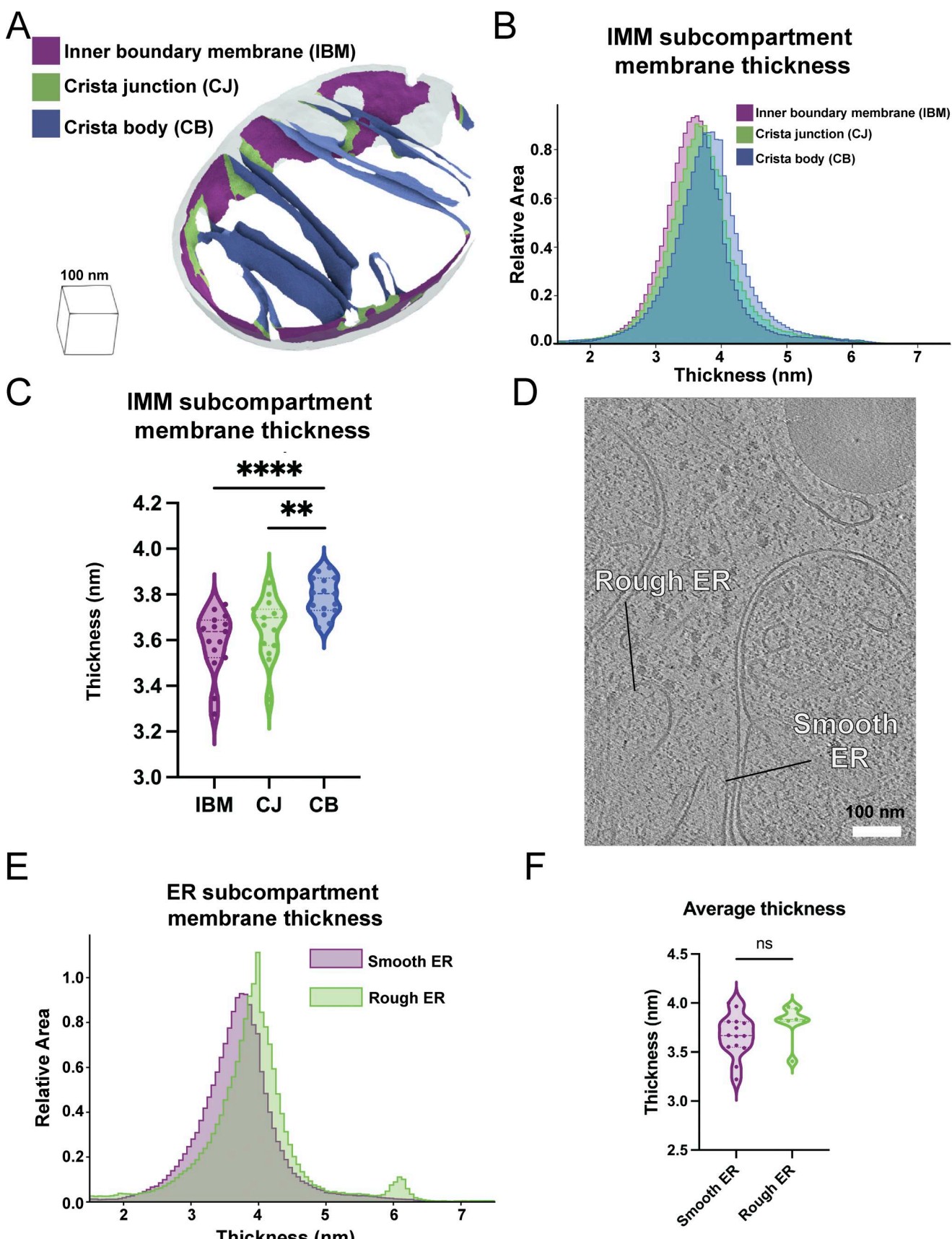

Figure 3. **Average membrane thickness varies within subcompartments of the same organelle. (A)** OMM distance-based classification automatically segments the IBM (purple), CJ (green), and CB (blue) in surface mesh models. **(B)** Area-weighted histogram of per-triangle thickness measurements shows

variation between different mitochondrial subcompartments. **(C)** Violin plot of per-tomogram median thickness shows statistically significant variations between mitochondrial subcompartments. IBM: $n$ = 15; CJ: $n$ = 15; CB: $n$ = 15. P values from the Mann–Whitney U test are indicated. **P < 0.01, ****P < 0.001. **(D)** Tomogram showing rough and smooth ER. Classification based on the presence or absence of bound ribosomes on the ER membrane, respectively. **(E)** Area-weighted histogram of per-triangle thickness measurement across ER subcompartment. **(F)** Violin plot of per-tomogram median thickness of each ER sub-compartment. Rough ER: $n$ = 7; smooth ER: $n$ = 14. CB, crista bodies; CJ, crista junctions.

that they often colocalize with large, membrane-embedded complexes (Fig. 6 B). Surprisingly, given our previous analysis, we observe ATP synthase localizing to thicker regions even in regions with low apparent curvature (Fig. 6 B). Strikingly, we detect several macromolecules localized to regions of thinner membrane (Fig. 6 B). Although it is challenging to unambiguously identify all these macromolecules, based on their size and localization within the crista body, a subset likely represents components of the oxidative phosphorylation machinery.

To further investigate the unique structural properties of these macromolecules, we extended our patch-based (Chang et al., 2025) density scans to encompass regions beyond the IMM, extending into the mitochondrial matrix. All line scans show the characteristic double Gaussian peaks of the IMM, with additional peak profiles observed at greater distances away from the IMM surface (Fig. 6 C). For ATP synthase, this corresponds to an additional broad peak spanning from 7 to 17 nm, corresponding to the diameter of the F1 catalytic domain. Comparing the average of several line scans of ATP synthase with those generated using randomized patches along the IMM caused this peak to disappear (Fig. 6 D). This suggests that this patch-based line scan approach can uniquely identify distinct signatures of macromolecules, paving the way for future applications aimed at subclassifying distinct membrane-associated molecules in cells.

## Discussion

The diversity of organellar membranes has been extensively studied through lipidomics (Jacquemyn et al., 2017; Cortie and Else, 2015), proteomics (Rezaul et al., 2005; Rath et al., 2021), light microscopy (Schroeder et al., 2019), and EM (West et al., 2011; Ding et al., 2012). However, recent advances in cryo-ET have enabled the direct visualization of these membranes within cells with higher resolution than ever before. We describe an automated and robust method to measure organellar membrane thickness directly from observed density in cellular cryo-electron tomograms. A substantial body of work has used contour-based density scanning in single-tilt images of *in vitro* membrane bilayers (Heberle et al., 2020); our method advances these measurements into three dimensions within cells by taking advantage of our previously developed tools to automatically model membranes as triangle meshes in tomograms (Barad et al., 2023). These surface meshes serve as a scaffold to perform hundreds of thousands of voxel-based density line scans across entire organellar membranes (Fig. 1).

A key advantage of this method is that it is agnostic to the approach used for generating the initial voxel-based segmentation, due to the combination of the surface mesh model and the direct sampling of the original tomogram for thickness measurements. Various techniques exist for isolating or

segmenting membranes from cryo-tomograms into binarized volumes that can subsequently be attributed to specific organelles. These range from watershed transforms (Volkmann, 2002) to computer vision-based methods such as tensor voting (e.g., TomoSegMemTV [Martinez-Sanchez et al., 2014], ColabSeg [Siggel et al., 2024]), and more recent approaches using 2D/3D U-net architectures (e.g., EMAN2 [Chen et al., 2017], MemBrain-Seg [Lamm et al., 2024, *Preprint*], DeepFinder [de Teresa-Trueba et al., 2023]). To address segmentation challenges posed by low signal-to-noise ratios, many of these methods prioritize voxel intensity connectivity to generate visually complete membrane segmentations. While appropriate for visualization, this can lead to variations in membrane width. The expanded Surface Morphometrics pipeline overcomes these limitations by directly performing voxel-based line scans on the underlying density visible in the tomographic data. Additionally, tomograms can be interchanged with different binning or additional postprocessing with denoising or contrast enhancement algorithms as long as the tomogram's relative dimensions remain consistent, allowing for rapid adaptation of workflows (Fig. S1).

We applied our method to analyze organellar membranes within both mammalian and yeast cell types, and identified statistically significant differences across various physiological membranes (Fig. 2). Although the nominal values differ, we observe similar trends in the relative thickness of the OMM and IMM across both mammalian and yeast cells, with the OMM consistently exhibiting lower thickness (Fig. 2). The IMM is one of the most protein-rich membranes in the cell, having been reported as containing 60–70% protein content by mass (Krebs et al., 1979). In contrast, the OMM is reported to have a much lower protein content—45% protein content by mass (Sperka-Gottlieb et al., 1988). In addition to protein differences, the OMM and IMM also have distinct lipid compositions, with the IMM being enriched in cardiolipin, promoting higher curvature and stabilizing large protein complexes (Ikon and Ryan, 2017), which may also contribute to the observed differences in membrane thickness within the same organelle.

In tomograms of yeast cell lamellae, we frequently captured a wider variety of organellar membrane types within a single field of view. This is likely due to the compact organization and smaller size of yeast cells, which often allow multiple cells and organelles to be imaged simultaneously within the same lamella. This feature provided us with an opportunity to explore thickness variations across a more diverse set of organellar membranes within tomograms of yeast cell lamellae. Within these data, we observe that the plasma membrane is the thickest (4.2 nm) when compared to other organellar membranes. Our findings agree with reported computational modeling of the plasma membrane, which is predicted to have a larger head-to-head distance (4.3–4.4 nm) (Monje-Galvan and Klauda, 2015). Room

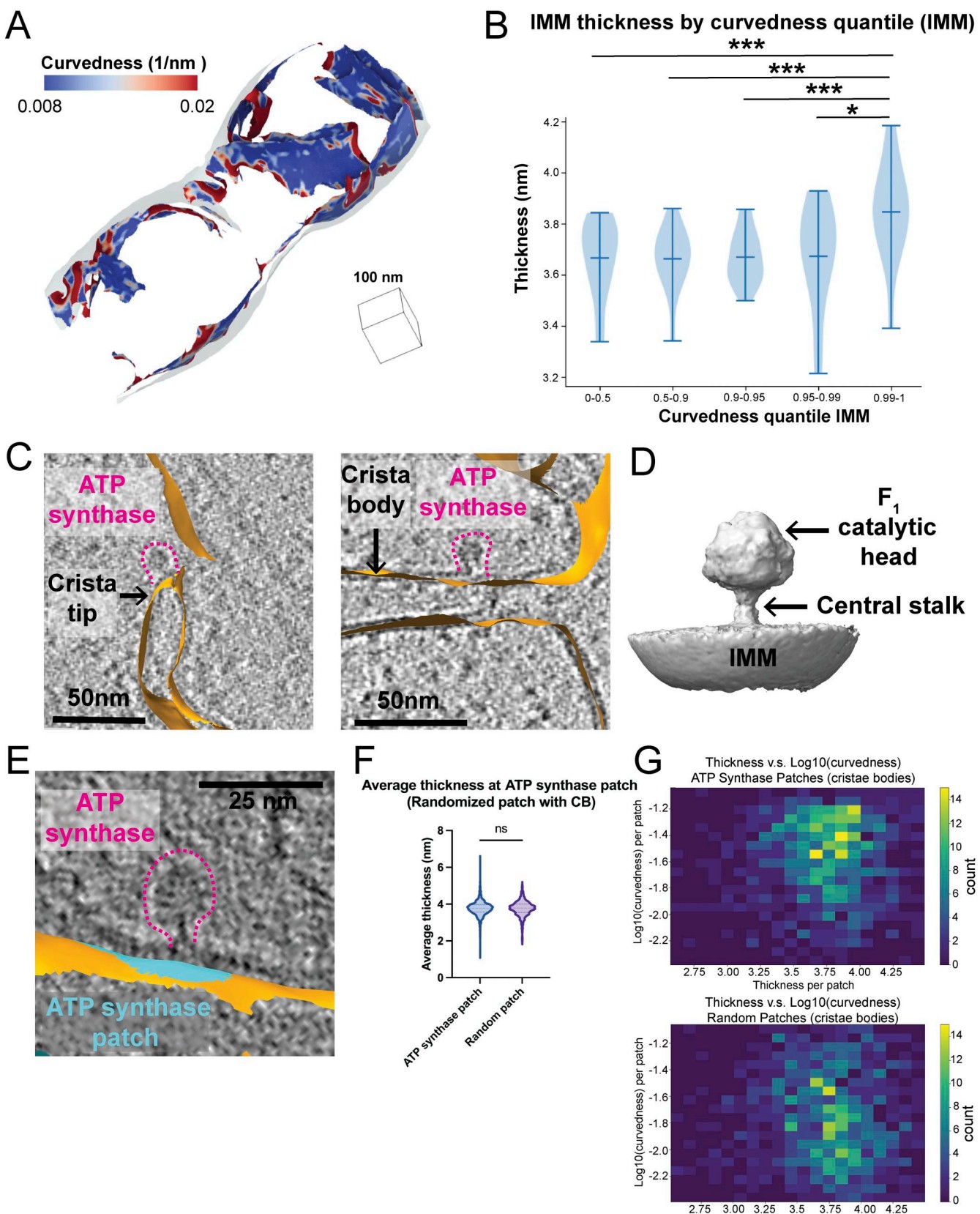

Figure 4. **Membrane thickness is positively correlated with membrane curvature. (A)** Representative membrane surface reconstruction of mitochondria colored by IMM curvedness. **(B)** Quantification of IMM thickness by curvedness quantile shows that higher curvature significantly correlates with thicker membrane. P values from the Mann–Whitney U test are indicated. *P < 0.05, ***P < 0.005. **(C).** ATP synthases (highlighted by pink dashed lines) are located at

the IMM (orange) with distinct curvature. The left panel shows an ATP synthase positioned at a curved crista tip, and the right panel shows one on a flat crista body. **(D)** 3D subtomogram average structure of ATP synthase monomer positioned on IMM with its F1 catalytic head and central stalk highlighted by arrows. **(E)** ATP synthase (outlined by pink dashed lines) and the corresponding membrane patch (cyan) representing its local footprint on IMM (orange). **(F)** Quantification of average thickness of ATP synthase patches and randomized patches on crista bodies for each mitochondrion. ATP synthase patches show no significant difference in thickness compared with randomized patches. Quantification of ATP synthase patch number $n = 984$ and randomized patch number $n = 984$ is shown. **(G)** 2D histogram of average thickness and $\log_{10}$ (curvedness) for ATP synthase patches and randomized patches on crista bodies. ATP synthase patches show a positive correlation between thickness and curvature.

---

temperature EM in concert with quantitative lipidomics of yeast organelles further supports this trend, showing that the plasma membrane has the thickest membrane and the highest level of ergosterol, which may contribute to its greater membrane thickness (Zinser et al., 1993; Schneiter et al., 1999). We measured the vacuole as the second thickest membrane, which is consistent with the room temperature EM data (Schneiter et al., 1999). The ability to directly measure membrane thickness in cells preserved in a frozen-hydrated state, without the use of chemical fixatives or stains, enables a more direct assessment of native membrane properties within the cellular context. Strikingly, we observe high variability in the thickness of intracellular vesicles. This could be because we are unable to distinguish between different vesicle types (which would typically need specialized CLEM approaches); therefore, we may be capturing thickness variations that reflect functional differences in vesicle subtypes that vary in both lipid content and membrane-associated cargo. Further studies of vesicles with specified origins, as well as with vesicle-originating organelles such as the plasma membrane and the Golgi apparatus, will help to differentiate these possibilities, as will studies of *in vitro* vesicles with defined lipid and protein composition.

Beyond differences in thickness between organelles, we detected local variation in thickness within physiologically distinct organelle subcompartments. In the IMM, we identified variation between the IBM, crista junctions, and crista bodies. These have sequentially increased thicknesses, with the crista body significantly thicker than either of the other two compartments (Fig. 3). Given the continuous membrane bilayer connecting these subcompartments, this local variation is likely due to a combination of local lipid enrichment and differences in protein content within the different subcompartments. In contrast, when comparing the differences between rough and smooth ER, the small apparent thickness difference was not statistically significant, though this may be due to the limited number of surfaces measured (smooth ER: $n = 14$ surfaces, rough ER: $n = 7$ surfaces). ER structures are diverse (Terasaki et al., 2013; Obara et al., 2023), and we anticipate that differences in thickness may often be found in physiologically distinct membrane subcompartments with local enrichments of both lipid and protein factors.

We found enhanced bilayer thickness in the most curved segments of the IMM (Fig. 4). This is in sharp contrast to *in vitro* biophysical studies, which reported that membrane resistance to curvature increases quadratically with bilayer thickness—the "thicker sandwich" is harder to bend (Bermúdez et al., 2004). This increase in thickness at the top 1% most curved triangles of the highest degrees of curvature is biophysically unfavorable, as thicker membranes are more rigid. This suggests that additional

forces, such as membrane-shaping proteins, may contribute to the increase in membrane thickness locally at these high-curvature regions. To address the possible sources of variation in thickness, we tested the measurements of thickness on tomograms collected on an *in vitro* vesicle sample, by comparing it with measurements made using existing techniques in two dimensions using cryo-EM from the same grid (Fig. 5). These data gave two major insights: first, our measurements with Surface Morphometrics varied by around 1.2% from the existing state-of-the-art method for measuring thickness in *in vitro* samples, suggesting that our technique can make both accurate and precise measurements of thickness in these conditions, as well as within cells. Second, we demonstrated that in these *in vitro* conditions, there is no correlation between thickness and curvature, suggesting that the difference we observe is a feature specific to the cellular environment, whether due to protein localization or the presence of specific curvature-inducing lipids such as cardiolipin. A limitation of this interpretation is that the very high curvatures we observed in cells were never observed with the vesicles—very small radius (8 nm or less) vesicles would be needed to observe such curvatures *in vitro*.

We reason that this difference from biophysical principles must be due to either specific lipids (cardiolipin, in particular, may increase both curvature and thickness together) (Golla et al., 2024) or membrane-bending proteins involved in generating that increased thickness. To address these possibilities, we attempted to evaluate the role of ATP synthase, one of the best characterized proteins involved in driving membrane curvature in cristae (Fig. 5). Consistent with its role, we show that regions of extremely high curvature in the IMM are associated with ATP synthase molecules. However, when comparing membrane regions in the presence and absence of ATP synthase, we did not identify any significant changes in membrane thickness. One potential explanation is that, within our dataset, we observe a mixture of ATP synthase monomers and dimers, and analyzing this heterogeneous population may obscure differences in membrane thickness.

Although ATP synthase did not significantly colocalize with membrane thickness, we discovered that many of the thickest and thinnest sections of membranes are associated with large membrane-associated protein complexes (Fig. 6). This aligns with our current understanding of protein–lipid interactions, which proposes that specific microdomains help stabilize and regulate protein complexes, such as the respiratory complexes (Friedman et al., 2015). This is suggestive that membrane bilayer variation may be a sensitive signature to help identify challenging-to-find membrane proteins within tomograms. In support of this concept, we extended our line scan approach beyond the lipid bilayer into the space around the membrane and

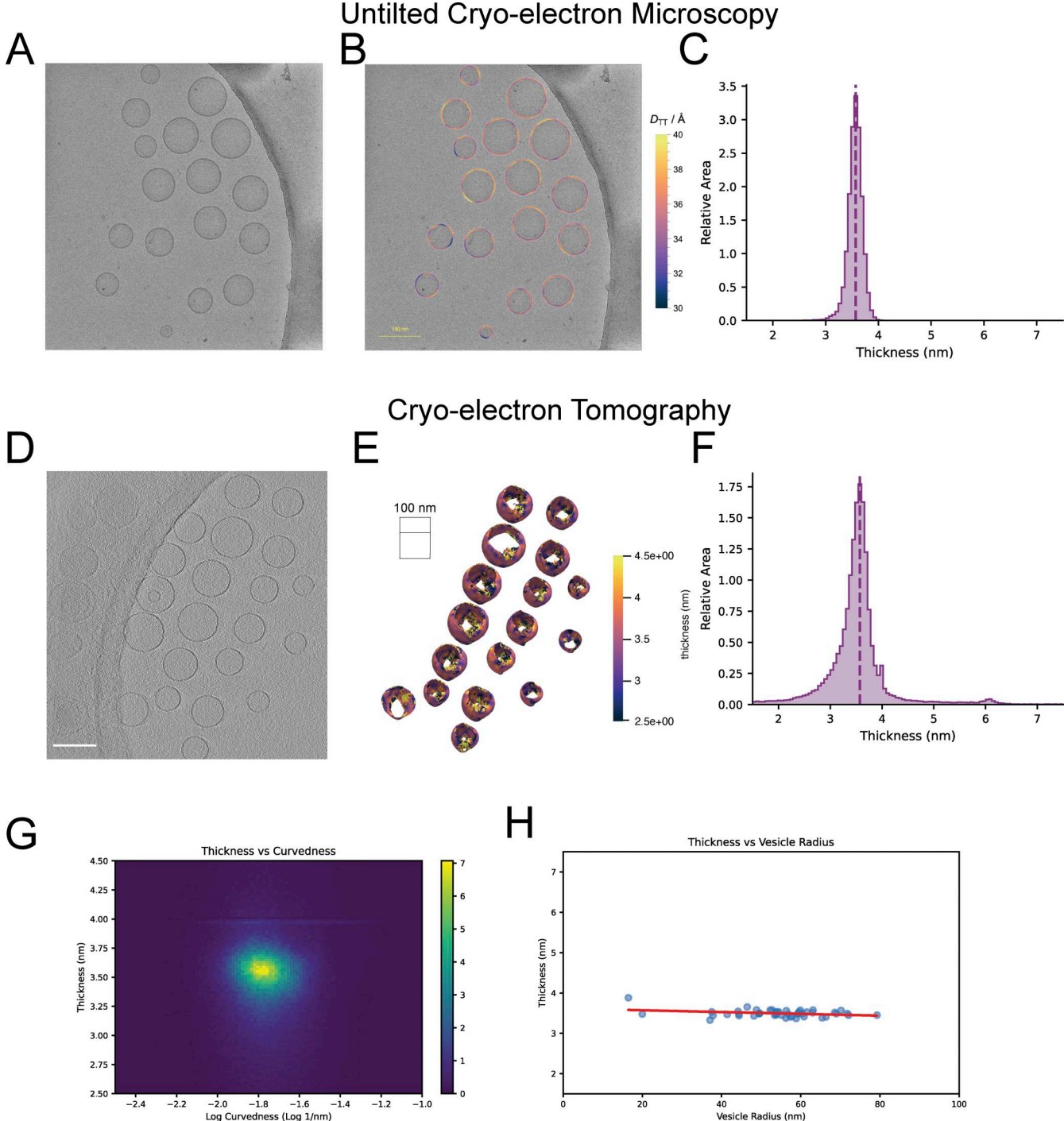

Figure 5.  **Thickness measurements of *in vitro* lipid bilayers match closely between cryo-EM and cryo-ET. (A)** Representative untilted 2D cryo-EM micrograph of *in vitro*–extruded vesicles with varying radii. **(B)** Thickness of vesicle bilayers, measured as previously described (Sharma et al., 2024), revealed in untilted micrographs, shown as color on the micrograph, reveals the thickness variation in the micrograph. The scale bar is 150 nm. **(C)** Histogram of thicknesses across contours in the micrographs showing the median thickness of 3.56 nm. **(D)** Representative slice of a reconstructed tomogram of *in vitro* vesicles acquired from the same grid. The scale bar is 150 nm. **(E)** Surface map shows the variation of thickness measured by the Surface Morphometrics pipeline. **(F)** Histogram of thickness across triangles in the tomograms shows the median thickness of 3.52 nm, <2% different than the *in vitro* micrographs, despite larger defocus and reduced signal-to-noise leading to increased variance across triangles. *n* = 3 tomograms, 47 vesicles. **(G)** 2D histogram showing no significant correlation between curvature and thickness in *in vitro* vesicles. **(H)** Average thickness per vesicle is plotted against vesicle radius, with a linear regression line in red, showing no significant correlation between vesicle radius and thickness. R$^2$ = 0.096.

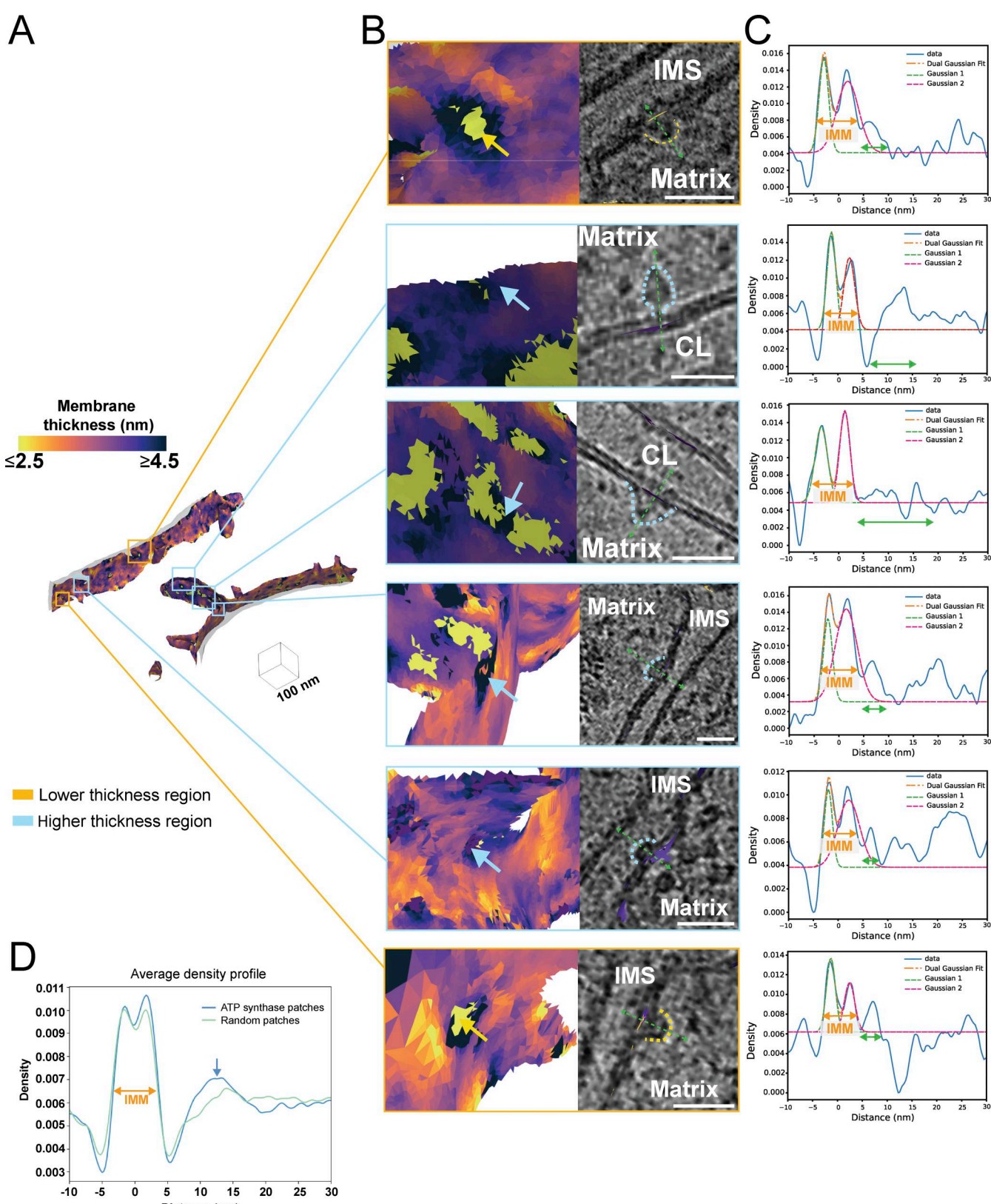

Figure 6. **Local variations in membrane thickness correspond to the presence of membrane-associated proteins. (A)** Representative membrane surface reconstruction of mitochondria colored by IMM membrane thickness. **(B)** Enlarged images of the corresponding boxed regions shown in A reveal the local membrane thickness and the colocalized membrane-associated macromolecules (outlined by dashed lines). Scale bars = 25 nm. **(C)** Line scan profiles along the green dashed arrow line in B of membrane-associated macromolecule patches show double Gaussian peaks (orange dashed line) corresponding to IMM (orange arrows), along with additional peaks representing protein macromolecules (green arrows). **(D)** Averaged line scan density profile across all ATP synthase patches (blue) shows a distinct peak compared with randomized patches (green).

showed that complexes like ATP synthase exhibit distinct line scan profiles. This opens the door to incorporating membrane mesh-guided density sampling as an assistive tool in protein localization and identification within tomograms.

In summary, our model-guided density-based approach to measuring membrane thickness reveals variation between organelles and within distinct organelle subcompartments, as well as within organelles in correlation with ultrastructural features such as curvature. Furthermore, this new and robust tool for measuring bilayer thickness has been incorporated into the Surface Morphometrics pipeline, enabling measurement of thickness from segmented tomograms in a fully automated manner in concert with other features such as membrane–membrane distances, curvature, and orientation. In this way, it will be straightforward for microscopists to measure the association of bilayer thickness with features such as membrane contact sites or curvature-separated subcompartments. We have already demonstrated the value of an early version of this approach in recent collaborative work studying ER-budded replication organelles in arbovirus-infected cells (Dahmane et al., 2024, Preprint). We look forward to seeing the discoveries made by other researchers using this new tool in the Surface Morphometrics pipeline.

## Materials and methods

### Preparation of vitrified MEFs on cryo-EM grids
MEFs expressing mitochondrially localized GFP (MEF$^{mtGFP}$) (Wang et al., 2012) were cultured in Dulbecco's modified Eagle's medium + GlutaMAX (Gibco) additionally supplemented with HiFBS (10%) and glutamine (4 mM) on fibronectin-treated (500 µg/ml, Corning) and UV-sterilized R ¼ Carbon 200-mesh gold EM grids (Quantifoil Micro Tools). After 15–18 h of culture, MEF$^{mtGFP}$ cells were plunge-frozen in a liquid ethane/propane mixture using a Vitrobot Mark 4 (Thermo Fisher Scientific). The Vitrobot was set to 37°C and 100% relative humidity, and blotting was performed manually from the back side of grids using Whatman #1 filter paper strips through the Vitrobot humidity/temperature chamber side port.

### Preparation of vitrified yeast (S. cerevisiae) cells
While most yeast data for this work have been previously published in EMPIAR-12534, we applied this analysis to an additional tomogram from the same dataset, which we have deposited as part of EMPIAR-13056. The preparation details are as follows: the yeast strain used in this study is a derivative of the S. cerevisiae strain BY4741, which contains Su9-mCherry-Ura3 and TIM50-GFP-His3MX6 (Lee et al., 2013). Yeast liquid cultures with an OD$_{600}$ of 0.8 were four times diluted to an OD$_{600}$ of 0.2 with the medium supplemented with 133 µg/ml CHX. The final concentration of CHX was 100 µg/ml. Yeast liquid cultures were incubated with CHX for 2 min, and then, 4 µl of the sample was applied to the glow-discharged R1/4 Carbon 200-mesh gold EM grid (Quantifoil Micro Tools). The EM grid was incubated in the chamber of Vitrobot (Vitrobot Mark 4; Thermo Fisher Scientific) for another 2 min before it was plunge-frozen in a liquid ethane/propane mixture. The Vitrobot

was set at 30°C with 100% relative humidity, and the blotting was performed manually from the back side of grids using Whatman #1 filter paper strips through the Vitrobot chamber side port.

### Cryo-fluorescence microscopy and mitochondrial network morphology scoring
Fluorescence and bright-field tiled image maps (atlases) of EM grids containing vitrified cellular samples that were acquired with a Leica CryoCLEM microscope (Leica DM6 Fixed Stage Fluorescence Microscope fit with a cryo-stage and an objective HC PL APO 50×/NA 0.9 CRYO CLEM) were collected using Leica LAS X software (25 µm Z stacks with system-optimized steps, GFP channel ex: 470, em: 525). Z stacks were stitched together in LAS X navigator to provide a single mosaic of maximum projections of the GFP signal, enabling rapid identification of the bulk mitochondrial morphology for each cell. For classification of mitochondrial network morphologies, max projections of individual tiles within fluorescence atlases of MEF$^{mtGFP}$ cells were randomized and blinded. Five researchers classified the cells as containing primarily elongated or fragmented mitochondria. Atlases were then exported from LAS X and loaded into MAPS software (Thermo Fisher Scientific) for fluorescence-guided milling.

### Fluorescence-guided milling of mouse embryonic fibroblasts
Cryo-FIB milling of lamellae was performed using an Aquilos dual-beam cryo-FIB/SEM instrument (Thermo Fisher Scientific) operated by xT software (Thermo Fisher Scientific). The fluorescence atlases were overlaid and aligned to SEM atlases of the same grid to target milling of MEF$^{mtGFP}$ cells with distinct mitochondrial network morphologies, as determined during blind classification (described above). MEF$^{mtGFP}$ cell targets were chosen based on their position within grid squares, the thickness of the ice in their vicinity, and their bulk morphology as assessed by the GFP fluorescence channel. Prior to milling, EM grids were coated with an organometallic platinum layer using a gas injection system for 3–4 s using an automation script (Barad, 2022), followed by a layer of platinum sputter. Targeted cells were milled using an automated cryo-preparation workflow (Buckley et al., 2020). Upon completion of final polishing of lamellae, an 8-s layer of platinum sputter was added to deposit platinum (bead) fiducials for downstream tilt series alignment. A total of two grids were milled for further tomography analysis.

### Fluorescence-guided milling of S. cerevisiae
Cryo-FIB milling of lamellae was performed using Aquilos 2 cryo-FIB/SEM (Thermo Fisher Scientific) operated by software xT, Maps, and AutoTEM (Thermo Fisher Scientific). Before milling, the EM grid was first subjected to a layer of platinum sputter for 15 s (1 kV, 20 mA, 10 Pa). Next, the grid was coated with an organometallic platinum layer using a gas injection system for 45 s and finally sputter-coated for 15 s (1 kV, 20 mA, 10 Pa). Automated milling was performed in AutoTEM using a previously detailed protocol (Chang et al., 2025). After the automatic milling and thinning process, a polishing step was manually executed using an ion beam of 50 pA and targeted for the thickness of lamella under 200 nm.

## Tilt series data collection

EM grids containing lamellae were transferred into a 300 keV Titan Krios microscope (Thermo Fisher Scientific), equipped with a K3 Summit direct electron detector camera (Gatan), and a BioQuantum energy filter (Gatan). Individual lamellae from both MEFs and *S. cerevisiae* were montaged with low dose (1 e⁻/Å²) at high magnification to localize cellular regions containing mitochondria, which were identified by their distinctive IMM and OMM. Data were collected to maximize the number of non-overlapping fields of view containing mitochondria, with no targeting of specific observed membrane ultrastructure. Tilt series were acquired using parallel cryo-ET (PACE-tomo) (Eisenstein et al., 2023), which is a set of Python-based SerialEM (Mastronarde, 2005) allowing multiple tilt series collection in parallel on the same lamella using beam shift. Tilt series were acquired at magnification 53,000× with a pixel size of 1.662 Å and a nominal defocus range between –4 and –6 μm. Data collection was done in a dose-symmetric scheme with 3° steps between –60° and +60° centered on –11° pretilt. Data were collected with dose fractionation, with 10 0.3001 e/Å² frames collected per second for MEF and 10~11 0.28–0.33 e/Å² frames for yeast. The total dose per tilt was ~3.0 e/Å², and the total accumulated dose for the tilt series was under 123 e/Å².

## Tilt series processing and reconstruction

Dose-fractionated tilt series micrograph movies underwent CTF estimation and motion correction in Warp (Tegunov and Cramer, 2019) and combined into averaged tilt series for alignment. Fractionated tilt series then were aligned using bead alignment using the post polish platinum fiducials in etomo (Mastronarde and Held, 2017). In some cases, the coverage of the platinum fiducials on the tilt series position was not amenable for bead tracking and patch tracking in etomo was used with 4 times binning and 36 binned pixel patches. Resulting contours were manually curated to remove poorly aligning patches, and the remaining contours were used for alignment and imported into Warp. Reconstruction was completed with Warp (Tegunov and Cramer, 2019) back-projection into 4, 6, and 8 times binned tomograms (corresponding to tomograms reconstructed at 6.65, 9.98, and 13.3 Å/pixel, respectively). Fiducial platinum "beads" were erased using a BoxNet model in Warp (Tegunov and Cramer, 2019). Tomogram thicknesses ranged from 80 to 150 nm.

## Membrane tracing, voxel segmentation, and surface generation

All reconstructed tomograms with eight and six times binning (voxel dimensions for MEF:13.30 × 13.30 × 13.30 Å; voxel dimensions for yeast: 9.98 × 9.98 × 9.98 Å) were processed by MemBrain-Seg (Lamm et al., 2024, *Preprint*), which is an advanced machine learning software based on U-Net architecture for tracing and segmenting cellular membranes. The binarized volumes of the traced membranes were then input into AMIRA (Thermo Fisher Scientific) for manual curation. All organellar membranes were designated as different labels using the 3D magic wand tool MEF (OMM, IMM, ER, smooth ER, rough ER, vesicles) and yeast (OMM, IMM, ER, nucleus, plasma membrane, vacuole, and vesicles). Manual clean-up of organellar membranes was performed using the 2D paintbrush tool. Final voxel segmentations were confirmed by visual inspection in AMIRA in comparison with the original tomogram. The voxel segmentation membrane label files were then exported from AMIRA and input into Surface Morphometrics (Barad et al., 2023). The labels of each membrane voxel segmentation were reconstructed as smooth surface meshes using the "segmentation_to_mesh.py." The surfaces were generated with a maximum of 200,000 triangles, a reconstruction depth of 8 (MEF) and 9 (yeast), and an extrapolation distance of 1.3 nm (MEF) and 1.5 nm (yeast). Curvature estimations of triangulated surface meshes were run using "run_pycurv.py." Inter- and intra-organelle distances were measured using "membrane_distance_orientation.py." Surfaces were subdivided into individual segments based on the connected components of the membrane graph to get "per-component" analyses to establish reasonable estimates of independent samples within each tomogram.

## Thickness measurement

For each triangle in a surface mesh, the density in the 9.98 Å/px tomogram (except in Fig. S1, where 6.65 and 13.30 Å/px were tested) was interpolated at along the normal vector at 0.25-nm increments ranging from 10 nm below the triangle to 10 nm above to produce a "line scan" revealing the electron density normal to the surface, revealing the areas of increased density corresponding to the head groups of each leaflet of the phospholipid bilayer. These line scans are generated in the "thickness_scan.py" script. These individual scans are very noisy due to the low signal-to-noise inherent to tomography data; to generate scans that could consistently fit, we applied two averaging strategies. For global thickness measurement, all line scans in a surface were averaged before fitting with a dual Gaussian distribution. For local measurements, each triangle's line scan was averaged with all neighboring triangles within 12 nm of the original triangle, with weighting that decreased with the distance from the central triangle by the following formula: $weight = \frac{1}{1+distance\,(nm)}$. In this way, closer triangles counted more for the assessment of thickness, minimizing the loss of local detail while enhancing signal-to-noise. With these averaged line scans, the PHG were modeled by fitting a pair of Gaussians to the line scan, with no correction applied for the white banding caused by defocus; despite this assumption, the fits are reasonably robust with an initial position guess estimated based on the peak position on each side of the midline. The measured thickness was determined by the difference in the means of the two-fit Gaussians. These thickness calculations were accomplished with the "thickness_plots.py" script. For each surface, the median thickness was calculated using weights corresponding to the area of each triangle, accounting for triangle size variation.

## Statistical inference

For all measurements, including thickness, curvature, and verticality, distributions were measured using a previously described area-weighted histogram technique to account for variance in the size of each segment being measured. For individual connected components, the area-weighted median of each

quantification was used as the overall measurement for that surface, to overcome issues with correlation of measurements between neighboring triangles causing overestimation of significance when per-triangle statistics are used. The mean and standard error–based 95% confidence interval of these per-surface measurements were reported, and all statistical comparisons of different surface types used the Mann–Whitney U test (Mann and Whitney, 1947), since in many cases the distributions of these measurements were visually non-normal. These statistics rely on standard tools in the "morphometrics_stats.py" component of the Surface Morphometrics toolbox and were generated in the "thickness_stats.py" script.

### Patch-based analysis

To analyze the local environment of ATP synthase, we defined ATP synthase-associated patches as regions on the IMM surface where ATP synthase particles localize (Blum et al., 2019). ATP synthase coordinates were obtained from the starfiles corresponding to each tomogram. The IMM surface coordinates and thickness for each mitochondrion were from triangle graph files (.gt) generated by the Surface Morphometrics pipeline. To identify patches for each mitochondrion, we first located the nearest IMM surface triangle to each ATP synthase particle using a k-dimensional tree Python function. To avoid cross-assignment ATP synthase from other mitochondria in the same tomogram, we excluded any nearest IMM triangles located farther than the height of an ATP synthase particle (24 nm). The remaining nearest IMM surface triangles were designated as "patch centers." Around each patch center, we searched for triangles within a 12-nm radius to define the ATP synthase–associated patch. Randomized ATP synthase–associated patches for each mitochondrion were generated based on the following criteria: (1) the number of randomized patches matched the number of ATP synthase–associated patches, and (2) the distances between randomized patch centers were >12 nm. This process was performed using "find_IMM_patches_for_ATP_synthase.py." The average thickness for each ATP synthase–associated patch and each randomized patch was calculated by "average_thickness_calculation_per_patch.py" and visualized as a violin plot. The Mann–Whitney U test was applied to assess the statistical significance of differences. The generation of the violin plot and statistical test was performed using Prism. The average curvature for each ATP synthase–associated patch and each randomized patch was calculated by "average_curvature_calculation_per_patch.py." The average thickness and $\log_{10}$ curvedness are plotted as a 2-dimensional histogram by "2dhist_curvedness_thickness.py."

To obtain the average ATP synthase line scanning density profile, we extracted ATP synthase–associated patches from multiple mitochondria as individual patch surfaces, preserving the coordinates and normal vectors of each surface triangle, using the script "extract_single_patch.py." Before performing line scanning, the normal vectors of the surface triangles in each patch were curated to ensure they pointed in the same direction as the vector from the patch center to the corresponding ATP synthase particle center. These curated patch surfaces were then correlated with the tomogram and served as the reference for the line scanning process. For each ATP synthase, a line scanning density profile was generated by sampling tomogram intensity values along the direction of the curated normal vectors. The scan extended from –10 nm (toward the inner membrane space) to +30 nm (toward the matrix) with 0.25 nm steps. The average ATP synthase line scanning density profile was then obtained by averaging the intensity profiles (IPs) across all ATP synthases. To obtain the line scanning profile for the other membrane-associated proteins, we picked the protein candidates as particles in ArtiaX and identified the corresponding patches with a 12-nm radius by the same approaches as ATP synthase. We performed line scans ranging from –10 to 30 nm on individual membrane-associated protein patches, using particle-curated vectors to obtain the density profile for each membrane-associated protein.

### Subcompartment analysis

Rough ER and smooth ER were separated on entire surfaces by visual inspection of the tomogram, identifying ribosome-bound ER membranes. For subcompartment analysis of IMMs, the inner membrane surface was subdivided based on additional distance from the OMM beyond the mode intermembrane distance measured for each surface, which corresponds to average spacing between the OMM and the IBM. All triangles <4 nm beyond this distance were classified as IBM. Triangles between 4 and 14 nm beyond this distance were classified as junctions. All triangles >4 nm beyond this distance were classified as crista bodies.

### *In vitro* comparative analysis

Phospholipids 1,2-dipalmitoyl-sn-glycero-3-phosphocholine (DPPC), 1-(12S-methylmyristoyl)-2-(13-methylmyristoyl)-sn-glycero-3-phosphocholine (a15-i15-PC), and 1-palmitoyl-2-oleoyl-sn-glycero-3-[phospho-rac-(1-glycerol)] (sodium salt) (POPG) were purchased from Avanti Polar Lipids. Cholesterol was purchased from Nu-Chek-Prep. All lipids were suspended in HPLC-grade chloroform and stored at –20°C until use. Concentrations of phospholipid stocks were determined using a colorimetric inorganic phosphate assay (Kingsley and Feigenson, 1979) (and the concentration of the cholesterol stock was determined gravimetrically). Large unilamellar vesicles (LUVs) composed of DPPC/a15-i15-PC/POPG/Chol (40/35/5/20 mol%) at a concentration of 3 mg/ml were prepared by first mixing the required volumes of lipid stocks in chloroform using a glass Hamilton syringe. The solvent was then evaporated using an inert gas stream, and the sample was kept under vacuum overnight. Dried lipid films were hydrated with ultrapure water preheated to 45°C and incubated for 1 h with vortex mixing every 15 min, followed by five freeze/thaw cycles between liquid nitrogen and a 45°C water bath. This suspension was then extruded 31 times through a 100-nm polycarbonate filter using a handheld mini-extruder (Avanti Polar Lipids) maintained at 45°C. The size and polydispersity of the LUVs were determined using dynamic light scattering (LiteSizer 100, Anton Paar U.S.A.) immediately after preparation and again before cryopreservation, which was performed 1 day after LUV preparation. Cryo-preservation was performed by adding 4 μl of LUVs to

a Quantifoil 2/2 carbon-coated 200-mesh copper grid (Electron Microscopy Sciences) that was glow-discharged for 30 s at 20 mA in a Pelco Easi-Glow discharge device (Ted Pella, Inc.). This was followed by manual blotting at room temperature, after which the grids were plunged into liquid ethane cooled with liquid nitrogen. The cryo-preserved grids were stored in liquid nitrogen.

### *In vitro* vesicle cryo-EM

Cryo-EM image collection was performed at ∼2 μm underfocus on a Titan Krios operated at 300 keV equipped with a Gatan K2 Summit direct electron detector operated in counting mode. Data collection was conducted in a semi-automated fashion using Serial EM software operated in low-dose mode. Briefly, areas of interest were identified visually, and 8 × 8 montages were collected at low magnification (2,400×) at various positions across the grid, with desired areas marked for automated data collection. Data were collected at 2.7 Å/pixel. Movies of 30 dose-fractionated frames were collected at each target site with the total electron dose being kept to <20 e−/Å². Dose-fractionated movies were drift-corrected with MotionCor2. Defocus and astigmatism were assessed with CTFFind4 (Rohou and Grigorieff, 2015, *Preprint*). Finally, a high-pass filter was applied along with phase flipping using the "mtffilter" and "ctfphaseflip" routines in the IMOD v4.11 software package (Mastronarde, 2024).

### *In vitro* vesicle tomography collection

Electron cryo-tomography data were collected on a different region of the same grid in attempts to minimize potential intergrid variability for comparisons with the LUV projection data. Regions of interest were identified as above, and points of acquisition for tilt series were placed across the grid square using Tomo5 (Thermo Fisher Scientific). Data were collected in super-resolution mode (1.36 Å/pixel) at tilt series ranging through ±52° at 3° tilt increments in dose-symmetric mode. Total electron dose was ∼ 80 e−/Å2. The data were acquired using a Gatan BioQuantum energy filter in zero-loss mode with a 20 eV filter on a K2 Summit camera in counting mode.

### 2D data thickness analysis

Projection images of vesicles were analyzed in Wolfram Mathematica v. 13 (Wolfram Research, Inc.) as previously described (Sharma et al., 2024) to obtain spatially resolved IPs in the direction normal to the bilayer. Briefly, vesicle contours (i.e., the set of points corresponding to the midplane of the projected bilayer as defined by a relatively bright central peak) were first generated using a neural network–based algorithm (MEMNET) that is part of the TARDIS software package (Kiewisz et al., 2024, *Preprint*). Vesicles meeting any of the following criteria were omitted from analysis due to the possibility of artifacts: (1) contact with the edge of the well; (2) location at the edge of the image; (3) containing internal debris, nested vesicles, or multiple lamellae; and (4) sufficiently nonspherical in shape. For each selected vesicle, the MEMNET contour was resampled at arc length intervals of 5 nm, resulting in a polygonal representation of the 2D contour. For each polygon, all pixels within a 5 nm × 20

nm rectangular region of interest centered at the face were selected, and their intensities were binned at 1 Å intervals in the long dimension (i.e., normal to the face) and subsequently averaged in the short dimension to produce a local segment IP. The local bilayer thickness, DTT, was calculated as the distance between the two minima of the local IP. Two methods were used to locate the minima: (1) a "model-free" method, in which a local 5-point Gaussian smoothing was first performed, and the distance between the two absolute minimum intensity values on either side of the central peak was determined; (2) a "model-fit" method that fits the profile as a sum of four Gaussians and a quadratic background, with the troughs corresponding to the two absolute minimum intensity values on either side of the central peak. The two methods typically agree to within 1 Å; the average of the two measurements was taken as the raw segment thickness. The final reported segment DTT values were obtained by local 4-point Gaussian smoothing of the raw segment thickness values.

### Subtomogram averaging of mitochondrial ATP synthase complexes

ATP synthase complexes were picked using a two-point directional picking strategy on a larger dataset of 43 tomograms from MEF^mtGFP cells that were either treated with degraded thapsigargin (500 nM, cat# 50-464-295; Thermo Fischer Scientific)—condition uncertain—or vehicle. Tomograms of binning 4 (6.65 Å) were postprocessed with a median filter in z direction (one iteration and kernel size of 11 pixels) and an mtf filter and used to manually particle pick prohibitin complexes using the software package i3 (Winkler, 2007). These particle picks were then converted to star files, and then, the warp2dynamo package was used to create dynamo tables (Burt et al., 2025). These particles were refined in dynamo for 5 cycles using the global refinement preset, and then, the rotation angles were randomized using *dynamo_table_randomize_azimut*h to avoid aligning all the particles on the missing-wedge artifact. This randomized table was then converted to star files using warp2dynamo package, and particles were extracted using Warp and refined in Relion (relion_autorefine) to achieve the resolution of 19 Å (Zivanov et al., 2022). Half-maps were postprocessed and refined in M and reached the resolution of 12.9 Å (Tegunov et al., 2021). ArtiaX module in ChimeraX was used to visualize these particles in the original tomograms (Pettersen et al., 2021; Ermel et al., 2022).

### Online supplemental material

Fig. S1 shows considerations for membrane thickness measurements in cryo-electron tomograms. Fig. S2 shows gallery of local thickness variations in organelles. Fig. S3 shows membrane thickness is not correlated with network morphology. Fig. S4 shows extreme thickness measurements reside in the mitochondrial cristae.

### Data availability

All tilt series, reconstructed tomograms, voxel segmentations, and reconstructed mesh surfaces used for quantifications were deposited in the EMPIAR under accession codes EMPIAR-13056. The tilt series and reconstructed tomograms from lamellae generated

from yeast cells were previously deposited under accession codes EMPIAR-12534. All subtomogram averages were deposited in the Electron Microscopy Data Bank (EMDB) under accession codes EMD-72321. The scripts used for patch analysis are available at https://github.com/GrotjahnLab/patch_analysis. All scripts used for Surface Morphometrics are available at https://github.com/grotjahnlab/surface_morphometrics.

## Acknowledgments

We thank Bill Anderson and William Lessin at The Scripps Research Institute Hazen cryo-EM facility and Venkata Mallampalli at the UTHSC-Houston structural biology core facility for microscope support. We thank Jean-Christophe Ducom and Lisa Dong at The Scripps Research Institute for computational support. We also thank R. Luke Wiseman, David DeRosier, Reika Watanabe, Kelsey C. Martin, and all other members of the Grotjahn Lab for their critical input on the manuscript.

M. Medina is supported by the ARCS (Achievement Rewards for College Scientists) Foundation. M. Frank is supported by the National Institute of Allergy and Infectious Disease (NIAID) grant 5T32AI170496-03. F.A. Heberle and M.N. Waxham are supported by NIH grant R01GM138887. M.N. Waxham acknowledges support from the Wiliam Wheless III Endowment. B.A. Barad is supported by the Collins Medical Trust. D.A. Grotjahn is supported by The Pew Scholars Program, Nadia's Gift Foundation Innovator Award of the Damon Runyon Cancer Foundation (DRR-65-21), and the National Institutes of Health (NIH) grant RF1NS125674. This work used equipment supported by NIH grant S10OD032467. Assistance from ChatGPT-4.0 (Open AI, https://chat.openai.com/) and Grammarly's AI tool was utilized to improve the clarity, grammar, and conciseness of the manuscript text.

Author contributions: Michaela Medina: conceptualization, data curation, formal analysis, investigation, methodology, validation, visualization, and writing—original draft, review, and editing. Ya-Ting Chang: conceptualization, formal analysis, investigation, methodology, software, validation, visualization, and writing—original draft, review, and editing. Hamidreza Rahmani: data curation, investigation, methodology, software, validation, and writing—original draft, review, and editing. Mark Frank: formal analysis, investigation, validation, and writing—review and editing. Zidan Khan: formal analysis, investigation, validation, and writing—review and editing. Daniel Fuentes: conceptualization, formal analysis, investigation, and methodology. Frederick A. Heberle: formal analysis, methodology, software, supervision, validation, visualization, and writing—review and editing. M. Neal Waxham: data curation, investigation, methodology, supervision, validation, visualization, and writing—review and editing. Benjamin A. Barad: conceptualization, formal analysis, investigation, methodology, project administration, software, supervision, validation, visualization, and writing—original draft, review, and editing. Danielle A. Grotjahn: conceptualization, funding acquisition, methodology, project administration, resources, supervision, validation, visualization, and writing—original draft, review, and editing.

Disclosures: The authors declare no competing interests exist.

Submitted: 9 May 2025

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

**Supplemental material**

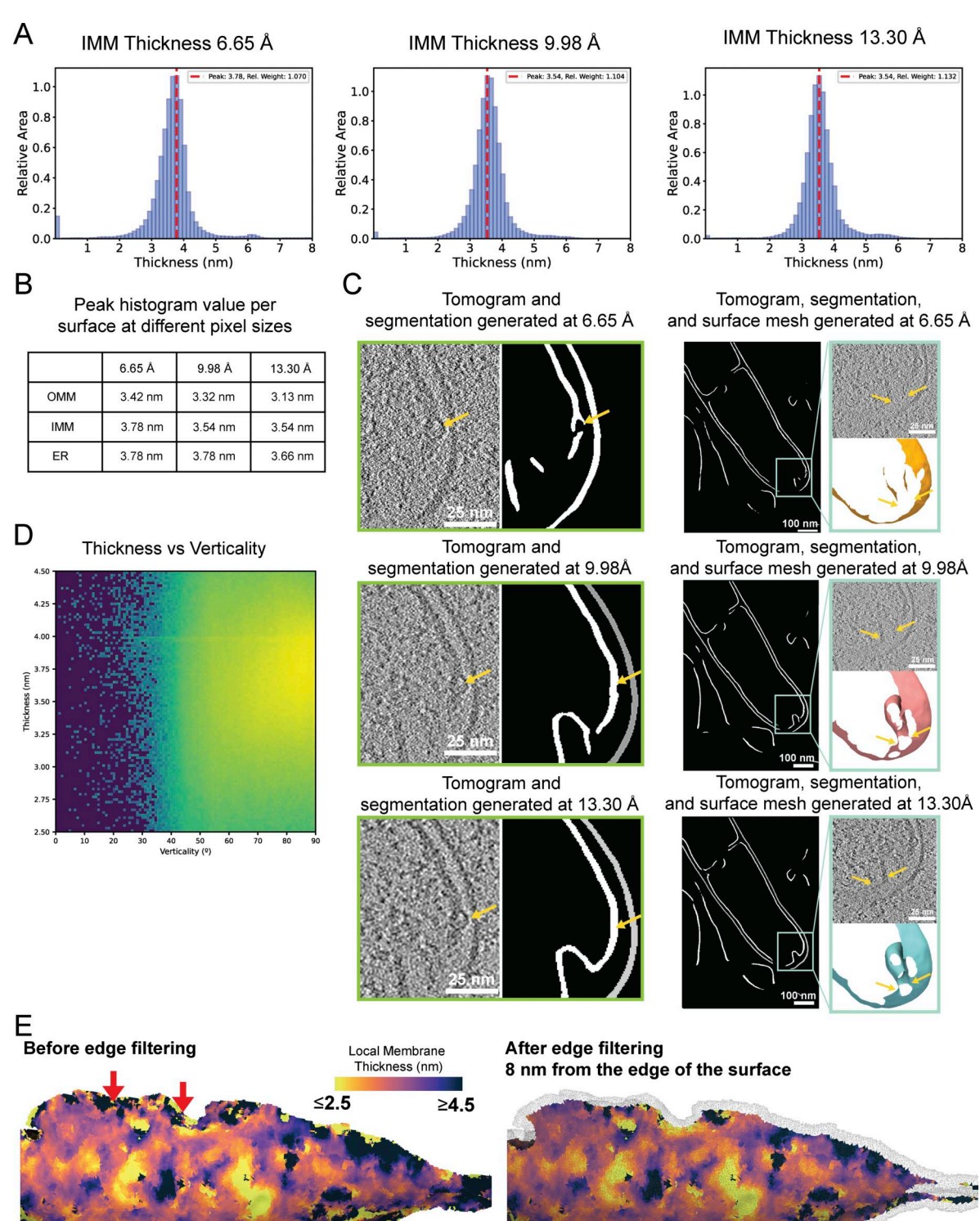

Figure S1.   **Considerations for membrane thickness measurements in cryo-electron tomograms. (A)** Area-weighted histograms of per-triangle thickness measurements of IMM thickness measurements performed on tomograms reconstructed at different pixel sizes. **(B)** Table of peak histogram values for membrane thickness calculations across distinct organellar membranes (OMM, IMM, and ER) performed on tomograms reconstructed at different pixel sizes. **(C)** Comparison of the resulting voxel segmentation output from MemBrain-Seg (Lamm et al., 2024, *Preprint*) performed on tomograms reconstructed at different pixel sizes 6.65, 9.98, and 13.3 Å/pixel. Voxel segmentations derived from tomograms reconstructed at 13.3 and 9.98 Å/pixel performed similarly accurately label the underlying membrane density visible in the tomogram. Voxel segmentations generated from tomograms reconstructed at 6.65 Å/pixel exhibit inaccuracies, such as voxels corresponding to membrane-protruding proteins mistakenly labeled as membrane. **(D)** 2D histogram of thickness measurement versus verticality of membrane shows no correlation between these measurements, suggesting that the missing wedge, which may artificially impact the thickness of membranes in these regions, is not driving significant errors in our data. **(E)** Triangulated surface mesh before edge filtering (left) and after edge filtering (right). The wire mesh shows the region of surface within 8 nm nearest to the edge of the membrane mesh that is artificially removed. Arrows highlight areas where measurements could not be made at all.

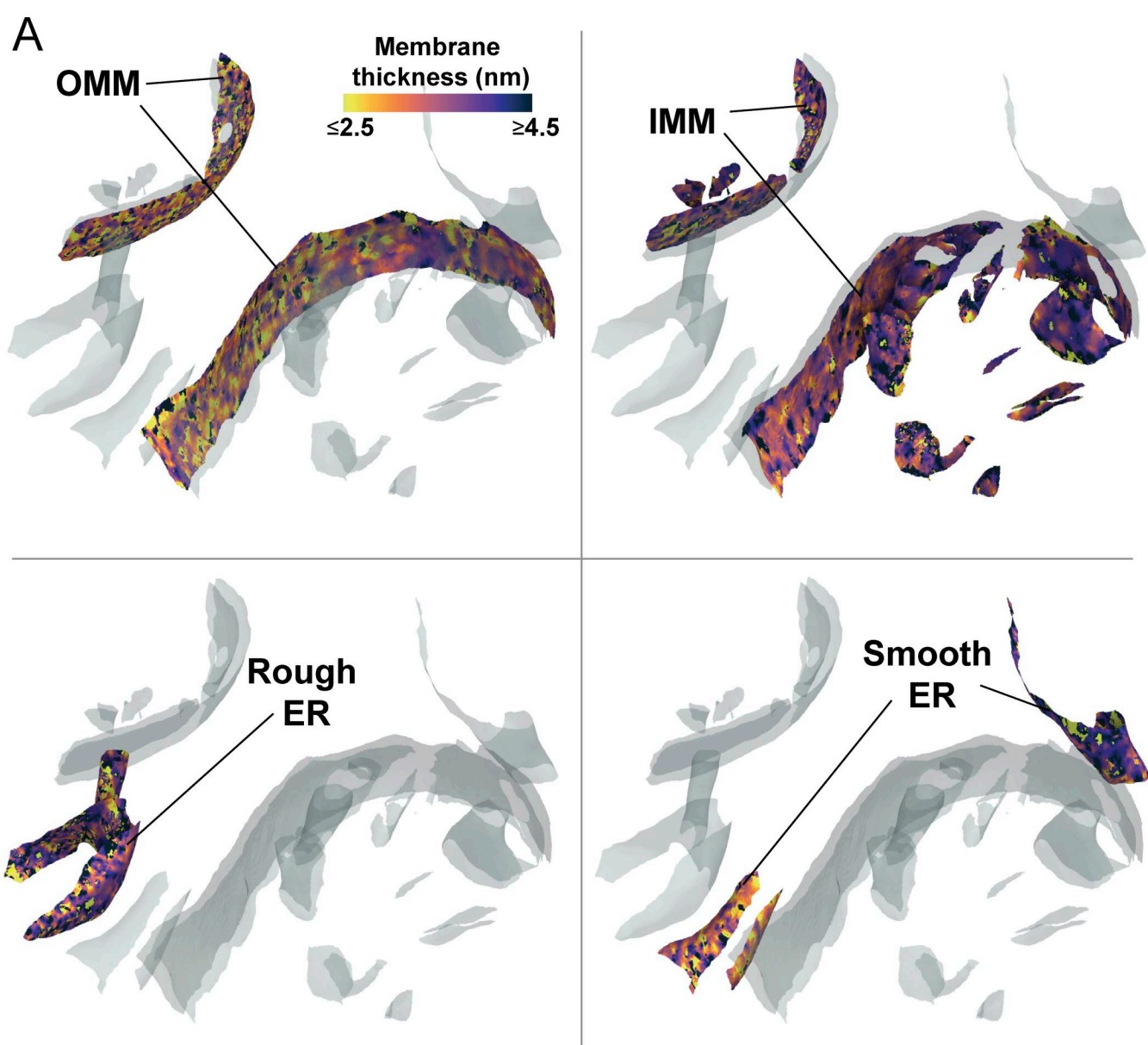

Figure S2. **Gallery of local thickness variations in organelles. (A)** Local thickness variations of individual physiological organelles are observed within a single tomogram, highlighting the variations both within and between organelles.

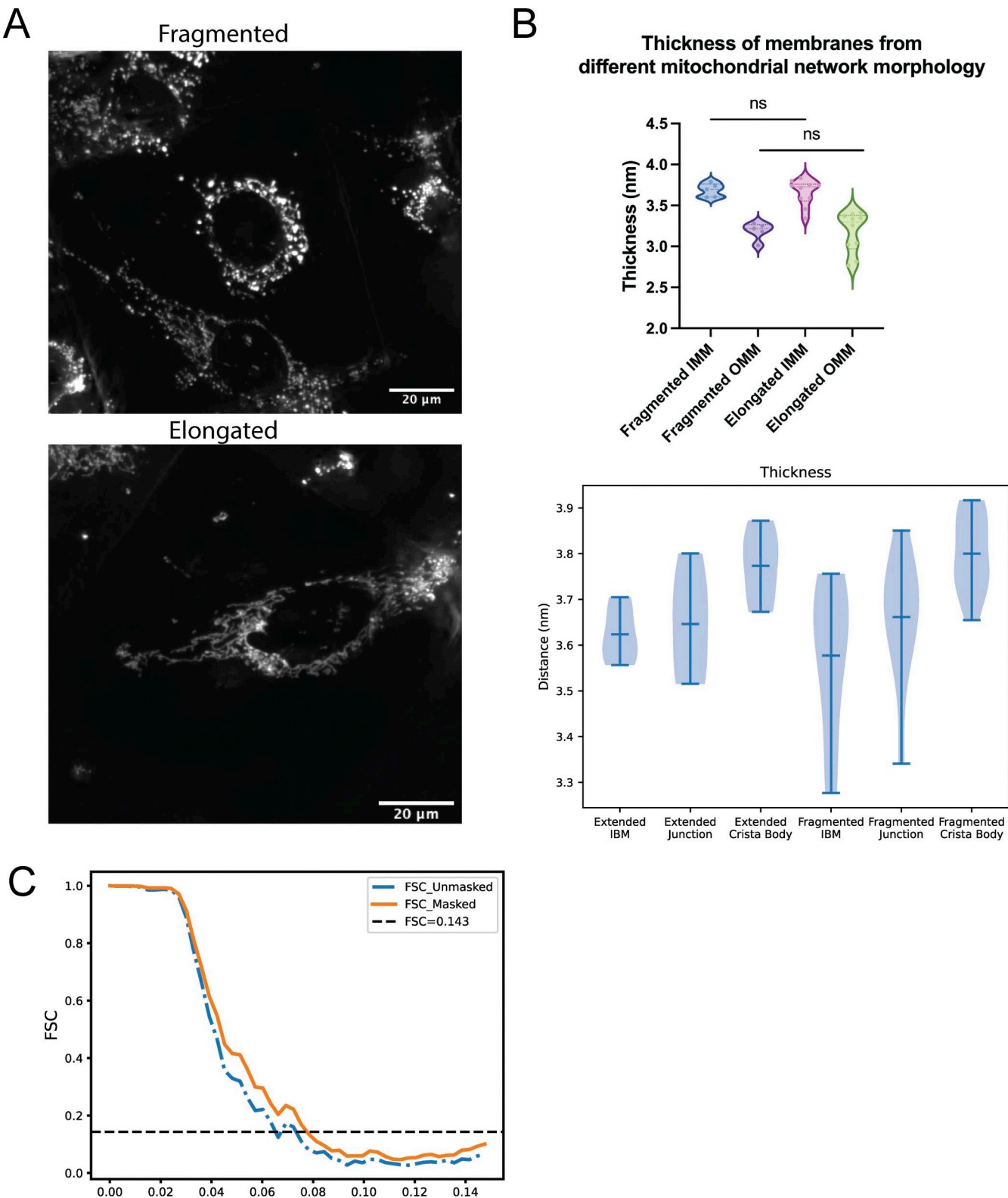

Figure S3. **Membrane thickness is not correlated with network morphology. (A)** Cryo-fluorescence microscopy of elongated and fragmented mitochondrial networks in MEF$^{mitoGFP}$ cells. **(B)** Violin plots displaying membrane thickness across different mitochondrial membranes and IMM subcompartments based on cellular mitochondrial network morphologies. **(C)** FSC curve of ATP synthase structure.

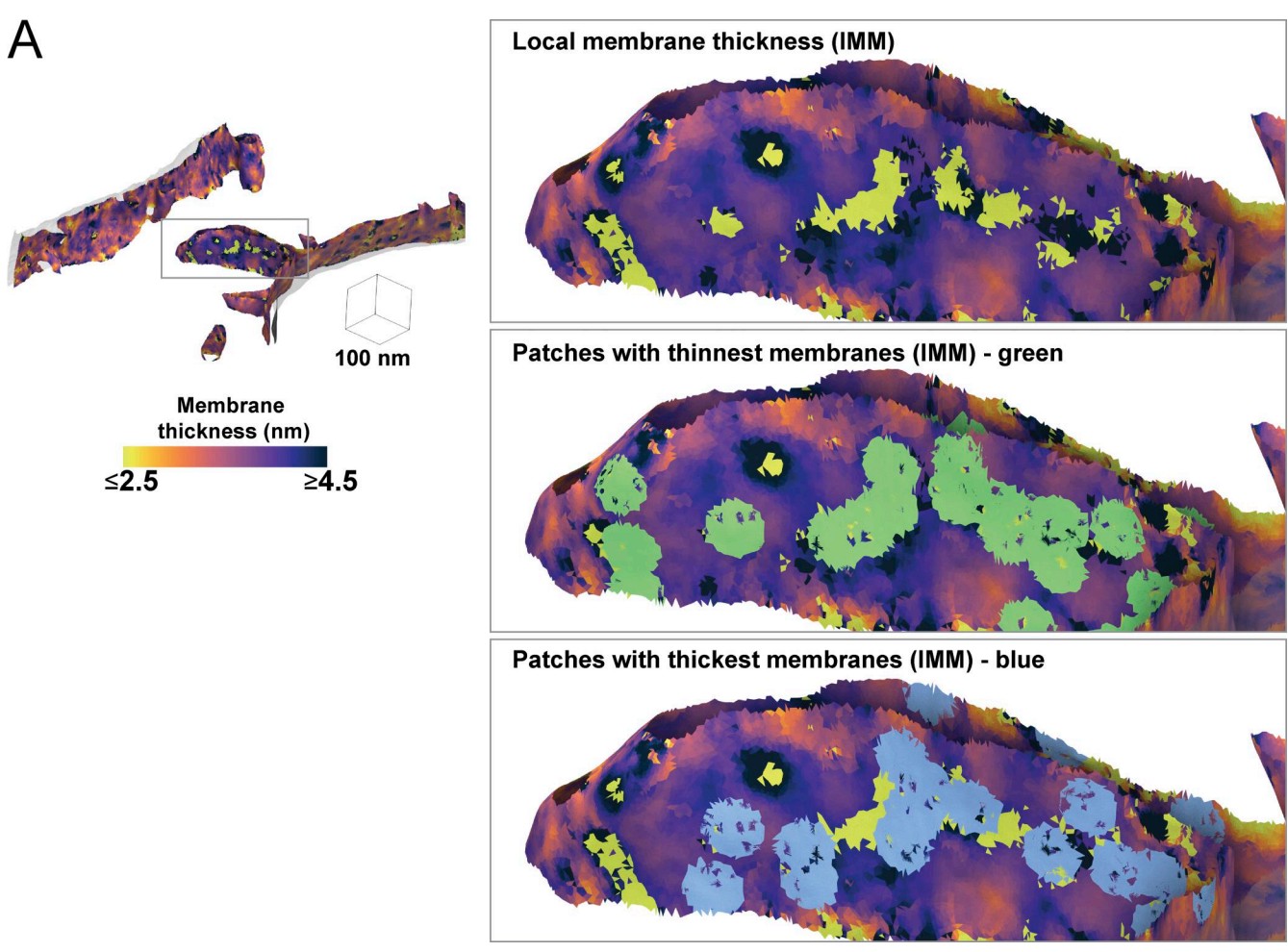

Figure S4. **Extreme thickness measurements reside in the mitochondrial cristae. (A)** Per-triangle local thickness variation within a tomogram can be used to generate patches with the 50 thinnest (green patches) and thickest (blue patches) triangles within each tomogram; these patches can be used for visually assessing protein content at these locations.

