## [Peer Review File · The Journal of Cell Biology]

Surface morphometrics reveals local membrane thickness variation in organelle subcompartments

Michaela Medina, Ya-Ting Chang, Hamidreza Rahmani, Mark Frank, Zidan Khan, Daniel Fuentes, Frederick Heberle, M Waxham, Benjamin Barad, and Danielle Grotjahn

Corresponding Author(s): Danielle Grotjahn, Scripps Research Institute

Review Timeline:

Submission Date:	2025-05-09
Editorial Decision:	2025-07-30
Revision Received:	2025-10-17
Editorial Decision:	2025-11-24
Revision Received:	2025-12-01

Monitoring Editor: Eva Nogales

Scientific Editor: Dan Simon

Transaction Report:

DOI: <https://doi.org/10.1083/jcb.202505059>

July 30, 2025

Re: JCB manuscript #202505059

Danielle Grotjahn
Scripps Research Institute

Dear Dr. Grotjahn,

Thank you for submitting your manuscript entitled "Surface morphometrics reveals local membrane thickness variation in organellar subcompartments." The manuscript was assessed by expert reviewers, whose comments are appended to this letter. Thank you for your patience with the peer review process. We invite you to submit a revision if you can address the reviewers' key concerns, as outlined here.

You will see that the reviewers feel that your work provides a potentially valuable new method. However, a major concern shared by both reviewers is whether the presence of membrane proteins affects the measurements of membrane thickness. This is a key issue that would need to be thoroughly addressed in a revision. The reviewers also ask for a more detailed explanation of the methodology, increased sample size and measurements of other membranes to provide a more rigorous assessment of the different cellular membranes, additional analyses to address possible effects of noise and the missing wedge effect, and improvements in data presentation and discussion.

GENERAL GUIDELINES:

Text limits: Character count for an Tools is < 40,000, not including spaces. Count includes title page, abstract, introduction, results, discussion, and acknowledgments. Count does not include materials and methods, figure legends, references, tables, or supplemental legends.

Figures: Tools may have up to 10 main text figures. Figures must be prepared according to the policies outlined in our Instructions to Authors, under Data Presentation, <https://jcb.rupress.org/site/misc/ifora.xhtml>. All figures in accepted manuscripts will be screened prior to publication.

Supplemental information: There are strict limits on the allowable amount of supplemental data. Tools may have up to 5 supplemental figures. Up to 10 supplemental videos or flash animations are allowed. A summary of all supplemental material should appear at the end of the Materials and methods section.

Please note that JCB now requires authors to submit Source Data used to generate figures containing gels and Western blots with all revised manuscripts. This Source Data consists of fully uncropped and unprocessed images for each gel/blot displayed in the main and supplemental figures. For assays performed using capillary electrophoresis and/or immunoassay-based detection, authors should instead provide the electropherogram graph(s) for each experiment, plotting fluorescence/chemiluminescence intensity vs. molecular weight/size. Please be sure to provide one Source Data file for each figure gels, blots, and/or capillary electrophoresis assays along with your revised manuscript files. File names for Source Data figures should be alphanumeric without any spaces or special characters (i.e., SourceDataF#, where F# refers to the associated main figure number or SourceDataFS# for those associated with Supplementary figures). For traditional gels and blots, the lanes of the gels/blots should be labeled as they are in the associated figure, the place where cropping was applied should be marked (with a box), and molecular weight/size standards should be labeled wherever possible. For capillary electrophoresis assays, each trace in the graph should be color-coded and labeled to indicate which protein, gene, or sample is being measured (please try to avoid red/green combinations to accommodate our color-blind readers).

The typical timeframe for revisions is three to four months. If you anticipate any difficulties in meeting this aforementioned revision time limit, please contact us and we can work with you to find an appropriate time frame for resubmission. Please note that papers are generally considered through only one revision cycle, so any revised manuscript will likely be either accepted or rejected.

Thank you for this interesting contribution to Journal of Cell Biology. You can contact us at the journal office with any questions at cellbio@rockefeller.edu.

Sincerely,

Eva Nogales, PhD
Monitoring Editor
Journal of Cell Biology

Dan Simon, PhD
Scientific Editor
Journal of Cell Biology

Reviewer #1 (Comments to the Authors (Required)):

Lipid bilayers form the structural foundation of organellar architecture and compartmentalization in cells. Previous biophysical, biochemical, and imaging studies, utilizing purified and reconstituted liposomes, have established that lipid composition critically influences membrane physical properties such as thickness and curvature. Here, the authors present a novel method for quantifying membrane thickness based on cryo-ET. This technique leverages triangulated surface mesh reconstructions to calculate membrane thickness using voxel density-based line scans across organellar membranes. Based on this approach, the authors reveal significant membrane thickness variations both within individual organelles and across distinct organellar membranes and a positive correlation between membrane thickness and curvedness. Although that these findings provide a framework to investigate how changes in membrane composition affect ultrastructure, the manuscript still needs to address several key concerns listed below:

Major concerns:

1. The authors utilized the distance between the phospholipid head groups (PHG) of the membrane as the readout of membrane thickness. The authors also mentioned in the manuscript, protein abundance varies across different membranes, and highly abundant membrane proteins can even form a protein layer-like structure. The authors also noted that both proteins and membranes contribute to electron density, which may affect the accuracy of segmentation and membrane thickness measurements. Therefore, the authors need to clarify whether protein signals were taken into account during the measurement of membrane thickness. Otherwise, the reliability of the membrane thickness analysis presented in the study could be compromised.
2. The authors found a positive correlation between membrane thickness and curvedness. Indeed, it is most probable that proteins incur higher membrane curvedness and then render membrane in a thicker state. To verify this, the authors need to prepare pure liposomes in varying diameters and measure their respective diameters via their methods. This will help authors to claim the underlying mechanism between membrane thickness and curvedness.
3. To reduce the computation burdens, the authors used tomograms with a pixel size of approximately 0.7 or 1.3 nm, which will inevitably compromise segmentation accuracy and consequently increase measurement errors. To validate the robustness of this computational workflow, the authors should employ smaller pixel sizes to reconstruct the tomograms, followed by instance membrane segmentation and membrane thickness measurements.
4. The authors measured the membrane thickness of mitochondria (IMM and OMM), ER, and vesicles after the cell milling. It will be more systematic to analyze the membrane thickness of the other major membrane systems including nuclear membrane and plasma membrane. It is more interesting to explore the membrane thickness of plasma membrane in the varying osmotic conditions.
5. The manuscript appears to have been prepared hastily. In the Data and Code Availability section, the EMPIAR and EMDDB accession codes are missing. Additionally, the provided GitHub link does not contain the relevant analysis scripts. These omissions make it difficult to fully evaluate the methodological details of the study.
6. In Fig2D and Fig3F, the sample numbers for vesicles, rough ER, and smooth ER are less than 5. For better statistical analysis, it is quite necessary to enlarge the dataset scale and re-analyze the p values.

Reviewer #2 (Comments to the Authors (Required)):

This manuscript by Medina et al. presents a new method to measure the lipid bilayer thickness based on cryo-ET reconstruction. I believe this is an important but under-explored topic. The authors proposed an approach using line scans across triangular surface meshes of segmented membranes to address this problem. They also employed a Gaussian mixture model to estimate the distances between the headgroups of the lipids. The manuscript presents several truly intriguing aspects of membrane thickness variations. For instance, they found that the membrane thickness distribution correlates with the local curvature. They showed that the membrane thickness changes around membrane-associated proteins, such as ATP synthases in mitochondria. Overall, the work is convincing and the questions they attempt to address are important to cell biology, especially the concept that employs high-resolution structural approaches to understanding cellular behaviors of membrane-bound organelles. I believe this represents something really at the interface of structural biology and cell biology. However, due to the high noise levels in conventional cryo-ET datasets, I would suggest that the authors provide more technical details and carefully revise their manuscript.

Here are some specific comments:

1. The authors discussed too many unnecessary, less relevant, or relatively standard procedures, while omitting lots of technical details of the key part of this work. For example, most of the routine procedures in cryo-ET sample preparation, data collection, parameter optimization can be moved to the supplementary or methods, and at the same time, the authors should provide sufficient assumptions, rationales, principles, statistical and computational details about their approach. In most cases, I only received information from the conceptual level, and it was impossible to judge whether one or another approach makes sense to this reviewer.

2. A recent study by Glushkova et al. (bioRxiv, 2025) reported partially consistent but notably different results using a different, yet conceptually similar, cryo-ET-based approach for systematic membrane thickness analysis. For instance, Medina et al. in this manuscript reports the inner mitochondrial membrane (IMM) thickness as 3.6 {plus minus} 0.07 nm and the outer mitochondrial membrane (OMM) as 3.2 {plus minus} 0.10 nm. By contrast, Glushkova et al. reported significantly higher ranges: IMM at 4.6-5.1 nm and OMM at 4.2-4.8 nm.

This level of discrepancy between the two studies far exceeds the reported differences between IMM and OMM within each study. I am worried that the divergence may arise not from biological variability but rather from technical or analytical inconsistencies in the two papers. This can be attributed to several possible factors that may include differences in segmentation strategy, resolution, fitting methodology (Euclidean 3D distances vs. 1D Gaussian fitting), or other post-processing steps. Regardless of all possibilities at the methodological level, I am more concerned about the intrinsic information one can ultimately retrieve from normal cryo-ET datasets due to the very high noise level, missing wedge effects and contributions from protein densities.

I strongly encourage the authors to comment on this discrepancy, and potentially address my concerns regarding the retrievable information from cryo-ET datasets.

3. In some of the figures, I found that the membrane thickness starts from 0 nm. I agree this may happen in noisy datasets as long as we employ any statistic approaches, but the results simply do not make sense to me.

4. Significant effects from missing wedge. It is well-known that cryo-ET reconstructions are limited by the missing wedge effect. I appreciate the authors attempt to address the membrane thickness problem, but the membrane segments may simply adopt different orientations during the cryo-ET reconstruction. This will lead to severe differences in the estimates of the membrane thicknesses. Thus, I suggest that the authors perform additional analysis to address this effect.

5. I wonder if the membrane thicknesses distributions may also be biased by existence of membrane proteins throughout the figures in this manuscript, such as Figure 2B, 5B, S1, S2 and S4. It seems to me that Figure 3D clearly shows protein densities that may widen the 'nominal' membrane thickness. I suspect that the statistics here might be simply results of biases from uncharacterized proteins. It would be very nice to perform a bit more careful analysis to avoid this possibility.

Response to Review

Editor Comments:

Thank you for submitting your manuscript entitled "Surface morphometrics reveals local membrane thickness variation in organellar subcompartments." The manuscript was assessed by expert reviewers, whose comments are appended to this letter. Thank you for your patience with the peer review process. We invite you to submit a revision if you can address the reviewers' key concerns, as outlined here.

You will see that the reviewers feel that your work provides a potentially valuable new method. However, a major concern shared by both reviewers is whether the presence of membrane proteins affects the measurements of membrane thickness. This is a key issue that would need to be thoroughly addressed in a revision. The reviewers also ask for a more detailed explanation of the methodology, increased sample size and measurements of other membranes to provide a more rigorous assessment of the different cellular membranes, additional analyses to address possible effects of noise and the missing wedge effect, and improvements in data presentation and discussion.

We would like to thank the reviewers and editors for their thoughtful comments and suggestions for improvement of this manuscript. We have now addressed all concerns below, which include the incorporation of new datasets, analysis, figures, and additions to the text. The details regarding how we addressed specific concerns are highlighted below.

Reviewer #1 (Comments to the Authors (Required)):

Lipid bilayers form the structural foundation of organellar architecture and compartmentalization in cells. Previous biophysical, biochemical, and imaging studies, utilizing purified and reconstituted liposomes, have established that lipid composition critically influences membrane physical properties such as thickness and curvature. Here, the authors present a novel method for quantifying membrane thickness based on cryo-ET. This technique leverages triangulated surface mesh reconstructions to calculate membrane thickness using voxel density-based line scans across organellar membranes. Based on this approach, the authors reveal significant membrane thickness variations both within individual organelles and across distinct organellar membranes and a positive correlation between membrane thickness and curvedness. Although that these findings provide a framework to investigate how changes in membrane composition affect ultrastructure, the manuscript still needs to address several key concerns listed below:

We thank this reviewer for their constructive feedback. We have incorporated the suggestions for improvement and performed additional experiments to address the reviewer's major concerns. Overall, we feel that this feedback substantially strengthened the revised manuscript and improved its clarity and significance.

Major concerns:

1. The authors utilized the distance between the phospholipid head groups (PHG) of the membrane as the readout of membrane thickness. The authors also mentioned in the manuscript, protein abundance varies across different membranes, and highly abundant membrane proteins can even form a protein layer-like structure. The authors also noted that both proteins and membranes contribute to electron density, which may affect the accuracy of segmentation and membrane thickness measurements. Therefore, the authors need to clarify whether protein signals were taken into account during the measurement of membrane thickness. Otherwise, the reliability of the membrane thickness analysis presented in the study could be compromised.

We thank the reviewer for bringing this important point to our attention. The surface mesh reconstructions generated using our Surface Morphometrics pipeline approximate the mid-surface/plane of the bilayer. This is then used as a geometric model for calculating voxel density line scans. Since the resulting individual line scans of our tomographic data are noisy, we incorporate weighted averaging using a specified radius and weight the measurements based on the distance from the center triangle. In preparation for this work, we tested various averaging radii and determined that an averaging radius of 12 nm yielded clear peaks representing the signal from the PHGs (**Figure 1D**). At this radius, it is clear that the predominant signal comes from the phospholipid head groups, and appears to dominate any signal that may result from single- or multi-pass transmembrane protein regions. This is evident in the ‘line scan signatures’ we generated in **Figure 6** at regions that contain visible membrane-associated proteins, such as ATP synthase. In the resulting line scans, we observe clear peaks corresponding to the phospholipid head groups, despite knowing that there are many transmembrane helices in these regions. Therefore, for measuring membrane thickness, we have determined that an averaging radius of 12 nm yields a clear signal for the PHGs, enabling accurate and robust fitting of a double Gaussian at these peaks. However, the radius of averaging is a parameter that can be fine-tuned, and we anticipate that lowering the radius of averaging may result in higher sensitivity to variations caused by transmembrane helices. These analyses are beyond the scope of the current manuscript, which aims to use these density scans to calculate local membrane thickness, but highlights the exciting potential of our approach to probe local structural properties of membranes from cryo-ET data.

To better illustrate the separation between the fitting of the membrane and the localization of protein density, we have significantly updated **Figure 6** (formerly **Figure 5**) to both highlight more examples of localized protein density (green double arrows) while clearly showing the fit of the bilayer density in a separate position from the protein density (orange double arrows). This point is also clarified in the text starting on page 10, line 25:

*“All line scans show the characteristic double Gaussian peaks of the IMM, with additional peak profiles observed at greater distances away from the IMM surface (**Figure 6C**).”*

2. The authors found a positive correlation between membrane thickness and curvedness. Indeed, it is most probable that proteins incur higher membrane curvedness and then render membrane in a thicker state. To verify this, the authors need to prepare pure liposomes in varying diameters and measure their respective diameters via their methods. This will help authors to claim the underlying mechanism between membrane thickness and curvedness.

We thank the reviewer for this excellent suggestion. We agree that this is an important experiment that would shed light on the underlying mechanisms governing the positive correlation between membrane thickness and curvedness observed in the cellular context, which we propose is likely mediated through the presence of membrane proteins.

To address this, we collaborated with the groups of Professors Fred Herberle and M. Neal Waxham, who previously developed an approach to measure membrane thickness from 2D cryo-EM projections of purified vesicles. They prepared and imaged an in vitro liposome sample, collecting both high-dose untitled 2D projection micrographs and low-dose tilt series images from the same sample deposited on the same electron microscopy grid. We then applied their approach to the 2D untitled cryo-EM micrographs and our Surface Morphometrics pipeline and thickness approach to the reconstructed 3D tomograms. Notably, we observed near-identical thickness measurements between the two approaches. Furthermore, we did not observe a correlation between curvedness and membrane thickness in this sample, which is made up solely of

phospholipids. Collectively, these data support our hypothesis that the presence of membrane-embedded proteins likely contributes to the positive correlation we observe in the cellular context.

We have revised this manuscript to include an additional figure (**Figure 5**) and text (starting on page 9, line 21 and page 13, line 22) highlighting these results. Since this resulted in considerable additional data collection, processing, and analysis, our revised manuscript lists Dr. Mark Frank, Mr. Zidan Khan, and Drs. Heberle and Waxham as contributing authors.

“Thickness measurements from the Surface Morphometrics pipeline on 3D reconstructed data closely match those from previous approaches on 2D data of an in vitro vesicle sample.

To determine the degree to which changes might be due to artifacts from tomography data collection and to benchmark our measurements against alternative imaging modalities, we performed thickness calculations on a sample of in vitro extruded vesicles from untilted (2D projections) cryo-EM micrographs using previously described approaches^{8,9} and in tomographic data (3D reconstructions) using our Surface Morphometrics pipeline (Figure 5). To ensure consistency across imaging and sample conditions, we collected these data from the same sample deposited on the same electron microscopy (EM) grid. Both methods revealed thickness variations between 3 and 4 nm, though more variation was measured in the tomographic data, likely due to increased defocus, artifacts from tomogram reconstruction, and reduced signal-to-noise (Figure 5B-E). Despite these differences in variation, the two techniques showed remarkably similar overall thickness measurements, with a median of 3.56 nm for cryo-EM and 3.52 nm for cryo-ET. This results in a difference of less than 2%, suggesting that the measurements made by the Surface Morphometrics pipeline are consistent with state-of-the-art techniques for measuring bilayer thickness using other imaging modalities. Because these in vitro vesicles contained no proteins, we used them to test the degree to which thickness and curvature correlate in the absence of protein factors (Figure 5G & H). We observed no statistically significant correlation between local membrane curvature and overall vesicle radius, suggesting that the increased thickness at high curvature in Figure 4G may be a specific feature of cellular membranes, possibly due to curvature-associated lipids or proteins.”

“To address the possible sources of variation in thickness, we tested the measurements of thickness on tomograms collected on an in vitro vesicle sample, comparing it with measurements made using existing techniques in two dimensions using cryo-electron microscopy from the same grid (Figure 5). This data gave two major insights: first, our measurements with surface morphometrics varied by around 1.2% from the existing state-of-the-art method for measuring thickness in in vitro samples, suggesting that our technique can make both accurate and precise measurements of thickness in these conditions, as well as within cells. Second, we demonstrated that in these in vitro conditions there is no correlation between thickness and curvature, suggesting that the difference we observe is a feature specific to the cellular environment, whether due to protein localization or the presence of specific curvature-inducing lipids such as cardiolipin. A limitation of this interpretation is that the very high curvatures we observed in cells were never observed with the vesicles – very small radius (8 nm or less) vesicles would be needed to observe such curvatures in vitro.”

3. To reduce the computation burdens, the authors used tomograms with a pixel size of approximately 0.7 or 1.3 nm, which will inevitably compromise segmentation accuracy and consequently increase measurement errors. To validate the robustness of this computational workflow, the authors should employ smaller pixel sizes to reconstruct the tomograms, followed by instance membrane segmentation and membrane thickness measurements.

In addition to increasing computational burden, we have found that segmentation performs poorly at lower binning (smaller pixel sizes), resulting in spurious inaccuracies in segmentation, particularly in regions with large, protruding membrane-associated densities. Furthermore, because our Surface Morphometrics workflow converts voxel data into continuous meshes, the resulting surfaces are intrinsically agnostic to pixel size. Once this voxel-to-surface conversion is performed, the surfaces can be rescaled to any desired pixel size without impacting the underlying geometry or resulting measurements.

Nonetheless, to systematically address this concern, we reconstructed a subset of tomograms at three distinct pixel sizes (6.65 Å, 9.98 Å, 13.3 Å) using our membrane segmentation (e.g., Membrain-2.0) and Surface Morphometrics thickness measurement pipeline (now shown in **Supplemental Figure 1**). We found that segmentations generated from tomograms at larger pixel sizes (i.e., 9.98 Å, 13.3 Å) were nearly identical and showed excellent overlap with the underlying membrane signal visible in the tomogram. However, consistent with our previous observations, we found that segmentations generated from tomograms at smaller pixel sizes contained many inaccuracies, as evidenced by the misclassification of membrane-protruding proteins as membrane and incomplete regions of membrane segmentation (**Supplemental Figure 1C**). When using a consistent surface mesh, the thickness measurements were relatively similar between all pixel sizes (**Supplemental Figure 1A, B**). To balance speed, accuracy, and resolution, we opted to use the tomograms reconstructed to 9.98 Å pixel size for our thickness calculations.

To address this in the text, we have significantly updated **Supplemental Figure 1** and these results in the text (starting on page 5, line 25) in the revised manuscript:

“To test the robustness of this approach to different data processing conditions, performed this analysis on a range of binned tomograms (6.65 Å/pixel, 9.98 Å/pixel, 13.30 Å/pixel) (Supplemental Figure 1A-C). We found that thickness variations We found that using binned tomograms (9.98 Å/pixel) yielded segmentation outputs that faithfully segmented the underlying membrane density visible in the tomogram, without artifacts due to protruding membrane proteins or blurring of the two membrane leaflets (Supplemental Figure 1C). Therefore, we concluded that, for this set of data, tomograms reconstructed to 9.98 Å/ Å/pixel provide the best balance between accuracy of segmentation and clear resolution of lipid bilayers. Upon inspection of the generated surfaces, we observed less reliable thickness measurements at the edges of the surface meshes due to artifacts caused by the missing wedge (Supplemental Figure 1D & E). To ensure that we only included membranes that had reliable thickness estimations, we implemented an edge exclusion feature to exclude all triangles within 8 nm of the edge of the original surface. This led to robust and accurate membrane thickness measurements that are free from artifacts inherent to cryo-ET data. All reported membrane thickness measurements were generated with edge filtering applied to the surface mesh reconstructions.”

4. The authors measured the membrane thickness of mitochondria (IMM and OMM), ER, and vesicles after the cell milling. It will be more systematic to analyze the membrane thickness of the other major membrane systems including nuclear membrane and plasma membrane. It is more interesting to explore the membrane thickness of plasma membrane in the varying osmotic conditions.

We agree that our analysis will open up numerous exciting new experimental avenues to explore changes in membrane thickness across various cellular membranes and physiological contexts. While we share the reviewer’s enthusiasm for exploring how membrane thickness varies under different osmotic conditions, we respectfully argue that it is beyond the scope of our paper to include multiple additional experimental groups, each requiring substantial investment in sample preparation and optimization. However, we agree that it is interesting to expand our analysis to explore thickness variations across other cellular membranes. Therefore,

we applied our surface morphometrics thickness measurement approach to our recently deposited data of tomograms generated from yeast cell (*Saccharomyces cerevisiae*) lamellae (EMPIAR-12534), which displays additional cellular membranes, including vacuole membrane, plasma membrane, various vesicles, and the nuclear envelope. Consistent with our analysis of mouse embryonic fibroblasts (MEFs), we observe significant differences in membrane thickness across all cellular membranes in our yeast dataset. We observe that similar trends in membrane thickness are consistent across these cell types, with the outer mitochondrial membrane exhibiting significantly thinner membrane relative to the inner mitochondrial and the endoplasmic reticulum membranes in both MEF and yeast cells.

We have added these results as additional panels in **Figure 2C-E** and text describing these results and discussion (starting on page 6, line 25 and page 12, line 13):

*“To further assess the performance of our pipeline across distinct organellar membranes and cell types, we calculated membrane thickness on tomograms of yeast (*Saccharomyces cerevisiae*) cell lamella with visible plasma membrane, vacuole membrane, and nuclear envelope bilayers from both new data and a previously published dataset (EMPIAR-12534)¹⁷. Consistent with our analysis in MEF cells, we detected significant differences in membrane thickness among various cellular membranes (Figure 2E-H). We observed a similar trend in membrane thickness variations in yeast as in MEF cells, with OMM (2.8 ± 0.27 nm) being significantly thinner than IMM (3.4 ± 0.13 nm), ER (3.8 ± 0.15 nm) and vesicles (4.1 ± 1 nm). In addition, we observe membrane thickness variations across the bilayers of the plasma membrane (4.2 ± 0.16 nm), vacuole membrane (4.1 ± 0.18 nm), and nuclear envelope (3.5 ± 0.23 nm), with the plasma membrane exhibiting the largest thickness values. Taken together, we show that different organelle membranes exhibit significant differences in average membrane thickness in multiple species, demonstrating the power of our approach to quantify subnanometer-level differences in membrane thickness across membranes visualized in their native environment.”*

*“In tomograms of yeast cell lamellae, we frequently captured a wider variety of organelle membrane types within a single field of view. This is likely due to the compact organization and smaller size of yeast cells, which often allow multiple cells and organelles to be imaged simultaneously within the same lamella. This feature provided us with an opportunity to explore thickness variations across a more diverse set of organellar membranes within tomograms of yeast cell lamellae. Within these data, we observe that the plasma membrane is the thickest (4.2 nm) when compared to other organelle membranes. Our findings agree with reported computational modeling of the plasma membrane, which is predicted to have a larger head-to-head distance (4.3~4.4 nm)³⁸. Room temperature electron microscopy in concert with quantitative lipidomics of yeast organelles further supports this trend, showing that the plasma membrane has the thickest membrane and the highest level of ergosterol, which may contribute to its greater membrane thickness^{39,40}. We measured the vacuole as the second thickest membrane, which is consistent with the room temperature electron microscopy data⁴⁰. The ability to directly measure membrane thickness in cells preserved in a frozen-hydrated state, without the use of chemical fixatives or stains, enables a more direct assessment of native membrane properties within the cellular context. Strikingly, we observe high variability in the thickness of intracellular vesicles. This could be because we are unable to distinguish between different vesicle types (which would typically need specialized CLEM approaches); therefore, we may be capturing thickness variations that reflect functional differences in vesicle subtypes that vary in both lipid content and membrane-associated cargo. Further studies of vesicles with specified origins, as well as with vesicle-originating organelles such as the plasma membrane and the Golgi apparatus, will help to differentiate these possibilities, as will studies of *in vitro* vesicles with defined lipid and protein composition.”*

5. The manuscript appears to have been prepared hastily. In the Data and Code Availability section, the EMPIAR and EMDB accession codes are missing. Additionally, the provided GitHub link does not contain the relevant analysis scripts. These omissions make it difficult to fully evaluate the methodological details of the study.

We appreciate the reviewer's feedback and have carefully revised the manuscript to include the EMPIAR (EMPIAR-13056) and EMDB (EMD-72321) accession codes. We also updated the complete set of scripts in the corresponding Github repository: https://github.com/GrotjahnLab/surface_morphometrics/

6. In Fig2D and Fig3F, the sample numbers for vesicles, rough ER, and smooth ER are less than 5. For better statistical analysis, it is quite necessary to enlarge the dataset scale and re-analyze the p values.

We used a statistical approach that is consistent with our previous work (Barad, Medina et al JCB, 2023), in which we performed extensive statistical analysis and benchmarking experiments on ultrastructural measurements (e.g., curvature, distance, and orientation) generated using our Surface Morphometrics approach. This relies on a likely overly conservative estimation of sample independence, with a single "measurement" per tomogram; to better represent the sample diversity we observed, we have updated our analysis to separate the connected components of each surface to be individually analyzed - this lead to sample sizes of 7 and 14 while not changing the statistical significance. We added this information to our methods section, starting on page 20, line 8:

"Surfaces were subdivided into individual segments based on the connected components of the membrane graph to get "per-component" analyses to establish reasonable estimates of independent samples within each tomogram."

And an additional section that further clarifies our statistical inference:

"Statistical Inference

For all measurements, including thickness, curvature, and verticality, distributions were measured using a previously described area-weighted histogram technique to account for variance in the size of each segment being measured. For individual connected components, the area-weighted median of each quantification was used as the overall measurement for that surface, to overcome issues with correlation of measurements between neighboring triangles causing overestimation of significance when per-triangle statistics are used. The mean and standard error-based 95% confidence interval of these per-surface measurements were reported and all statistical comparisons of different surface types used the Mann-Whitney U-test⁵⁴, since in many cases the distributions of these measurements were visually non-normal. These statistics rely on standard tools in the `morphometrics_stats.py` component of the surface morphometrics toolbox and were generated in the `thickness_stats.py` script."

Moreover, we have now included additional data on yeast cells, which show similar trends, highlighting the robustness and accuracy of our statistical analysis framework.

Reviewer #2 (Comments to the Authors (Required)):

This manuscript by Medina et al. presents a new method to measure the lipid bilayer thickness based on cryo-ET reconstruction. I believe this is an important but under-explored topic. The authors proposed an approach using line scans across triangular surface meshes of segmented membranes to address this problem. They

also employed a Gaussian mixture model to estimate the distances between the headgroups of the lipids. The manuscript presents several truly intriguing aspects of membrane thickness variations. For instance, they found that the membrane thickness distribution correlates with the local curvature. They showed that the membrane thickness changes around membrane-associated proteins, such as ATP synthases in mitochondria. Overall, the work is convincing and the questions they attempt to address are important to cell biology, especially the concept that employs high-resolution structural approaches to understanding cellular behaviors of membrane-bound organelles. I believe this represents something really at the interface of structural biology and cell biology. However, due to the high noise levels in conventional cryo-ET datasets, I would suggest that the authors provide more technical details and carefully revise their manuscript.

We thank this reviewer for their enthusiasm regarding the significance and relevance of our approach in addressing an important area of biology. While we have been very careful to include robust statistical verification as described in our previous JCB paper originally describing the surface morphometrics toolkit (doi: [10.1083/jcb.202204093](https://doi.org/10.1083/jcb.202204093)), it is true that in this work we are pushing much further into the limits of resolution of cryo-electron tomography, and we have endeavored to both expand our controls and to benchmark our work against alternative approaches using *in vitro* vesicle data. We believe this effort has significantly strengthened the robustness of our toolkit and will be quite valuable for our future work.

Here are some specific comments:

1. The authors discussed too many unnecessary, less relevant, or relatively standard procedures, while omitting lots of technical details of the key part of this work. For example, most of the routine procedures in cryo-ET sample preparation, data collection, parameter optimization can be moved to the supplementary or methods, and at the same time, the authors should provide sufficient assumptions, rationales, principles, statistical and computational details about their approach. In most cases, I only received information from the conceptual level, and it was impossible to judge whether one or another approach makes sense to this reviewer.

Per the reviewer's suggestion, we have significantly reorganized the first section of the results where we describe thickness measurements and have moved many of the experimental details to the supplement, while others are more clearly explained - the remaining sample preparation and microscopy details are highlighted as important because they are known to be critical for resolution of leaflets. See text starting on page 4, line 24.

At the same time, we added an additional section that outlines the statistical framework and computational details of our approach. We also added a figure panel that outlines a graphical representation of the basic framework of our approach (**Figure 1A**). Additional text outlining the statistical analysis can be found starting on page 21, lines 25:

“Statistical Inference

For all measurements, including thickness, curvature, and verticality, distributions were measured using a previously described area-weighted histogram technique to account for variance in the size of each segment being measured. For individual connected components, the area-weighted median of each quantification was used as the overall measurement for that surface, to overcome issues with correlation of measurements between neighboring triangles causing overestimation of significance when per-triangle statistics are used. The mean and standard error-based 95% confidence interval of these per-surface measurements were reported and all statistical comparisons of different surface types used the Mann-Whitney U-test⁵⁴, since in many cases the distributions of these measurements were visually non-normal. These statistics rely on standard tools in the

``morphometrics_stats.py`` component of the surface morphometrics toolbox and were generated in the ``thickness_stats.py`` script.”

2. A recent study by Glushkova et al. (bioRxiv, 2025) reported partially consistent but notably different results using a different, yet conceptually similar, cryo-ET-based approach for systematic membrane thickness analysis. For instance, Medina et al. in this manuscript reports the inner mitochondrial membrane (IMM) thickness as 3.6 ± 0.07 nm and the outer mitochondrial membrane (OMM) as 3.2 ± 0.10 nm. By contrast, Glushkova et al. reported significantly higher ranges: IMM at 4.6-5.1 nm and OMM at 4.2-4.8 nm.

This level of discrepancy between the two studies far exceeds the reported differences between IMM and OMM within each study. I am worried that the divergence may arise not from biological variability but rather from technical or analytical inconsistencies in the two papers. This can be attributed to several possible factors that may include differences in segmentation strategy, resolution, fitting methodology (Euclidean 3D distances vs. 1D Gaussian fitting), or other post-processing steps.

Regardless of all possibilities at the methodological level, I am more concerned about the intrinsic information one can ultimately retrieve from normal cryo-ET datasets due to the very high noise level, missing wedge effects and contributions from protein densities.

I strongly encourage the authors to comment on this discrepancy, and potentially address my concerns regarding the retrievable information from cryo-ET datasets.

The reviewer raises reasonable concerns about the comparability of the approaches from Glushkova et al (bioRxiv, 2025) as well as our technique. Because one technique measures the thickness of a segmentation, while the other uses the tomogram itself to measure the distance between lipid headgroups, our approaches are fundamentally impossible to directly compare; many transformations happen during the segmentation process, most importantly, a final binarization step that uses a heuristically determined threshold and directly impacts the thickness of the segmentation. This does not make their technique invalid; the measurement of the segmentation thickness is computationally much simpler and robust to signal-to-noise issues, albeit at the cost of being sensitive to artifacts in the segmentation process, and they were self-consistent with their binarization thresholding in their work, so we do not believe any bias is introduced. It does, however, mean that the measurements between the two techniques cannot be reasonably compared, though we note that the relative differences we observe in our early figures match fairly reasonably with theirs.

With that said, we chose to address this concern in part by incorporating our new **Figure 5** and its associated results section (starting on page 9 lines 21 and page 13, line 22), where we collaborated with the Waxham and Heberle labs, who have previously measured bilayer thickness directly in *in vitro* vesicles using untilted cryo-electron microscopy. We collected tomograms and untilted micrographs of vesicles on the same grid in a single data collection, then performed the like-to-like comparison of thicknesses with their approach. We had much greater variance in the tilted data (**Figure 5F**) than theirs (**Figure 5C**), but the median of our measurements generally lay within 1.2% of theirs, under half an Angstrom difference. We believe this firmly establishes the accuracy of our method for measuring the true spacing between phospholipid head groups and the comparability of our technique with the existing state-of-the-art tools for *in vitro* data.

“Thickness measurements from the Surface Morphometrics pipeline on 3D reconstructed data closely match those from previous approaches on 2D data of an *in vitro* vesicle sample.

To determine the degree to which changes might be due to artifacts from tomography data collection and to benchmark our measurements against alternative imaging modalities, we performed thickness calculations on a sample of *in vitro* extruded vesicles from untilted (2D projections) cryo-EM micrographs using previously described approaches^{8,9} and in tomographic data (3D reconstructions) using our Surface Morphometrics

pipeline (Figure 5). To ensure consistency across imaging and sample conditions, we collected these data from the same sample deposited on the same electron microscopy (EM) grid. Both methods revealed thickness variations between 3 and 4 nm, though more variation was measured in the tomographic data, likely due to increased defocus, artifacts from tomogram reconstruction, and reduced signal-to-noise (Figure 5B-E). Despite these differences in variation, the two techniques showed remarkably similar overall thickness measurements, with a median of 3.56 nm for cryo-EM and 3.52 nm for cryo-ET. This results in a difference of less than 2%, suggesting that the measurements made by the Surface Morphometrics pipeline are consistent with state-of-the-art techniques for measuring bilayer thickness using other imaging modalities. Because these *in vitro* vesicles contained no proteins, we used them to test the degree to which thickness and curvature correlate in the absence of protein factors (Figure 5G & H). We observed no statistically significant correlation between local membrane curvature and overall vesicle radius, suggesting that the increased thickness at high curvature in Figure 4G may be a specific feature of cellular membranes, possibly due to curvature-associated lipids or proteins.”

*“To address the possible sources of variation in thickness, we tested the measurements of thickness on tomograms collected on an *in vitro* vesicle sample, comparing it with measurements made using existing techniques in two dimensions using cryo-electron microscopy from the same grid (Figure 5). This data gave two major insights: first, our measurements with surface morphometrics varied by around 1.2% from the existing state-of-the-art method for measuring thickness in *in vitro* samples, suggesting that our technique can make both accurate and precise measurements of thickness in these conditions, as well as within cells. Second, we demonstrated that in these *in vitro* conditions there is no correlation between thickness and curvature, suggesting that the difference we observe is a feature specific to the cellular environment, whether due to protein localization or the presence of specific curvature-inducing lipids such as cardiolipin. A limitation of this interpretation is that the very high curvatures we observed in cells were never observed with the vesicles – very small radius (8 nm or less) vesicles would be needed to observe such curvatures *in vitro*.”*

We do share extensive concerns about the limit of information available in cryo-ET data, although we would point out that the localization precision (which is a better equivalent to our thickness measurement than resolution) for high-quality subtomogram averaging structures has often been considerably better than 1 Å C-alpha RMSD. To test the specific effect of the missing wedge, we plotted the verticality (our previously defined measure of the “tilt” of the membrane) against the thickness for our membranes. We generally found no correlation, shown in **Supplemental Figure 5D**. However, this is likely because we are generally conservative about segmenting only very well supported membranes and have very little membrane mesh at all that would fall into the 0-30° missing wedge region - we do expect more “completed” segmentation (such as closing vesicles completely) to have large errors in thickness measurement, but we also do not generally trust any of the other membrane details in the missing wedge and have not incorporated that information in any previous work for the Surface Morphometrics pipeline.

We also used the same **Supplemental Figure 5** to test the effect of different binnings on both segmentation and our measurements, and showed that at very high binning (>13Å/px) our results were systematically smaller (due to blurring between the two leaflets), but that the measurements were fairly consistent between 6.67 and 9.98 Å/px, while we got much better segmentation at 9.98Å/px. We addressed the concerns about protein density influencing results earlier, but between the *in vitro* data and our line scans showing the separation between the leaflet density peaks and the membrane-associated protein density in **Figure 6**, we feel that our method is clearly visually separating protein from lipid density. With that said, we have begun future efforts to expand this line scan approach towards more intentional work identifying and classifying

membrane-associated proteins, and look forward to sharing that with the community in future work. We also address this in text starting on page 5, line 25 in the revised manuscript:

“To test the robustness of this approach to different data processing conditions, performed this analysis on a range of binned tomograms (6.65 Å/pixel, 9.98 Å/pixel, 13.30 Å/pixel) (Supplemental Figure 1A-C). We found that thickness variations We found that using binned tomograms (9.98 Å/pixel) yielded segmentation outputs that faithfully segmented the underlying membrane density visible in the tomogram, without artifacts due to protruding membrane proteins or blurring of the two membrane leaflets (Supplemental Figure 1C). Therefore, we concluded that, for this set of data, tomograms reconstructed to 9.98 Å/ Å/pixel provide the best balance between accuracy of segmentation and clear resolution of lipid bilayers. Upon inspection of the generated surfaces, we observed less reliable thickness measurements at the edges of the surface meshes due to artifacts caused by the missing wedge (Supplemental Figure 1D & E). To ensure that we only included membranes that had reliable thickness estimations, we implemented an edge exclusion feature to exclude all triangles within 8 nm of the edge of the original surface. This led to robust and accurate membrane thickness measurements that are free from artifacts inherent to cryo-ET data. All reported membrane thickness measurements were generated with edge filtering applied to the surface mesh reconstructions.”

3. In some of the figures, I found that the membrane thickness starts from 0 nm. I agree this may happen in noisy datasets as long as we employ any statistic approaches, but the results simply do not make sense to me.

We appreciate the reviewer's careful attention in pointing out this potential source of confusion regarding the limits of our approach. There are several scenarios in which the thickness calculation may “fail”, including instances of local membrane regions with poor resolution in the tomogram, where the phospholipid head groups of each bilayer are ‘blurred’ together and not individually distinguishable. In this case, the calculated thickness measurement is 0. We opted to reveal this measurement rather than discarding it so that the user has a way to visually and intuitively evaluate the underlying surface measurement model relative to the corresponding tomogram density (much like evaluating the fit of an atomic model to a cryo-EM map). Nevertheless, in the cumulative, bulk analyses (e.g., histograms), these values will appear unless discarded from the data (for example, in **Figure 2C&G**).

To more clearly outline the caveats and potential failure points of our approach, we have added a new supplemental figure (**Supplemental Figure 1**) to help users critically evaluate the resulting measurements and their biological relevance. We also added additional text in the methods section starting on page 21, line 1:

“In cases where the two leaflets are not separated, this method reports a head group distance of zero, with both gaussians in a single peak. These can be filtered or highlighted as needed during downstream processing.”

4. Significant effects from missing wedge. It is well-known that cryo-ET reconstructions are limited by the missing wedge effect. I appreciate the authors attempt to address the membrane thickness problem, but the membrane segments may simply adopt different orientations during the cryo-ET reconstruction. This will lead to severe differences in the estimates of the membrane thicknesses. Thus, I suggest that the authors perform additional analysis to address this effect.

Artifacts arising from the missing wedge are indeed inherent to all tomograms and should be carefully considered in any downstream data analysis workflow. To minimize their impact, we performed segmentation on weighted back-projected tomograms rather than reconstructions generated by iterative algorithms such as

SIRT, which can artificially extend membranes in the 'z' direction of the tomogram. Furthermore, as noted by the reviewer, we attempt to mitigate the potential impact of these artifacts on our thickness measurements by applying edge filtering, which excludes membrane regions located near the boundaries of the missing wedge (i.e., at the outer edges of the reconstructed membrane surfaces).

The author raises a point that the missing wedge may lead to membrane segments adopting different orientations. In our experience, the effect of the missing wedge is signal smearing and loss for membrane regions aligned with the missing wedge, rather than a systematic change in their apparent orientation. Nonetheless, to evaluate whether orientation could bias thickness measurements, we plotted membrane thickness as a function of orientation (i.e., verticality). This analysis revealed no correlation, suggesting that the missing wedge does not introduce a measurable orientation-dependent bias in our calculated membrane thickness values. This figure is now included in **Supplemental Figure 1D** and in text starting on page 6 line 4:

“Upon inspection of the generated surfaces, we observed less reliable thickness measurements at the edges of the surface meshes due to artifacts caused by the missing wedge (Supplemental Figure 1D & E).”

5. I wonder if the membrane thicknesses distributions may also be biased by existence of membrane proteins throughout the figures in this manuscript, such as Figure 2B, 5B, S1, S2 and S4. It seems to me that Figure 3D clearly shows protein densities that may widen the 'nominal' membrane thickness. I suspect that the statistics here might be simply results of biases from uncharacterized proteins. It would be very nice to perform a bit more careful analysis to avoid this possibility.

We thank the reviewer for raising this important point, which was also noted in point #1 by Reviewer 1. We addressed this in part by our benchmarking work (previously mentioned and described in **Figure 5**) with protein-free *in vitro* vesicles, showing that our results closely match the existing tools used for these systems with high-dose 2D projection cryo-electron microscopy data, suggesting that our measurements are consistent in the absence of protein with existing tools. Additionally, as discussed in our response to Reviewer 1, we show that the density due to membrane-associated proteins is generally distinct from the bilayer density, sufficiently so that the measurements of the bilayer are consistent. We've updated **Figure 6** (formerly **Figure 5**) to highlight the localization of the bilayer fits in the line scan in the presence of protein density to further clarify this. This point is also clarified in the text starting on page 10, line 25:

“All line scans show the characteristic double Gaussian peaks of the IMM, with additional peak profiles observed at greater distances away from the IMM surface (Figure 6C).”

November 24, 2025

RE: JCB Manuscript #202505059R

Danielle Grotjahn
Scripps Research Institute

Dear Dr. Grotjahn,

Thank you for submitting your revised manuscript entitled "Surface morphometrics reveals local membrane thickness variation in organellar subcompartments." We would be happy to publish your paper in JCB pending the minor changes recommended by the reviewers and final revisions necessary to meet our formatting guidelines (see details below).

A. MANUSCRIPT ORGANIZATION AND FORMATTING:

1) Text limits: Character count for Tools is < 40,000, not including spaces. Count includes title page, abstract, introduction, results, discussion, and acknowledgments. Count does not include materials and methods, figure legends, references, tables, or supplemental legends.

2) Figure formatting: Tools may have up to 10 main text figures. Scale bars must be present on all microscopy images, including inset magnifications. Please add scale bars to figures 1B & 6B.

Also, please avoid pairing red and green for images and graphs to ensure legibility for color-blind readers. If red and green are paired for images, please ensure that the particular red and green hues used in micrographs are distinctive with any of the colorblind types. If not, please modify colors accordingly or provide separate images of the individual channels.

3) Statistical analysis: Error bars on graphic representations of numerical data must be clearly described in the figure legend. The number of independent data points (n) represented in a graph must be indicated in the legend. Please indicate whether 'n' refers to technical or biological replicates (i.e. number of analyzed cells, samples or animals, number of independent experiments). If independent experiments with multiple biological replicates have been performed, we recommend using distribution-reproducibility SuperPlots (please see Lord et al., JCB 2020) to better display the distribution of the entire dataset, and report statistics (such as means, error bars, and P values) that address the reproducibility of the findings.

Statistical methods should be explained in full in the materials and methods. For figures presenting pooled data the statistical measure should be defined in the figure legends. Please also be sure to indicate the statistical tests used in each of your experiments (both in the figure legend itself and in a separate methods section) as well as the parameters of the test (for example, if you ran a t-test, please indicate if it was one- or two-sided, etc.). Also, if you used parametric tests, please indicate if the data distribution was tested for normality (and if so, how). If not, you must state something to the effect that "Data distribution was assumed to be normal but this was not formally tested."

4) Materials and methods: Should be comprehensive and not simply reference a previous publication for details on how an experiment was performed. Please provide full descriptions (at least in brief) in the text for readers who may not have access to referenced manuscripts. The text should not refer to methods "...as previously described."

5) For all cell lines, vectors, strains, constructs/cDNAs, etc. - all genetic material: please include database / vendor ID (e.g. Addgene, ATCC, etc.) or if unavailable, please briefly describe their basic genetic features, even if described in other published work or gifted to you by other investigators (and provide references where appropriate). Please be sure to provide the sequences for all of your oligos: primers, si/shRNA, RNAi, gRNAs, etc. in the materials and methods. You must also indicate in the methods the source, species, and catalog numbers/vendor identifiers (where appropriate) for all of your antibodies, including secondary. If antibodies are not commercial, please add a reference citation if possible.

6) Microscope image acquisition: The following information must be provided about the acquisition and processing of images:

- a. Make and model of microscope
- b. Type, magnification, and numerical aperture of the objective lenses
- c. Temperature
- d. Imaging medium
- e. Fluorochromes

f. Camera make and model

g. Acquisition software

h. Any software used for image processing subsequent to data acquisition. Please include details and types of operations involved (e.g., type of deconvolution, 3D reconstitutions, surface or volume rendering, gamma adjustments, etc.).

7) References: There is no limit to the number of references cited in a manuscript. References should be cited parenthetically in the text by author and year of publication. Abbreviate the names of journals according to PubMed.

8) Supplemental materials: Tools may have up to 5 supplemental figures and 10 videos. Please also note that tables, like figures, should be provided as individual, editable files. A summary of all supplemental material should appear at the end of the Materials and methods section. Please include one brief sentence per item.

9) eTOC summary: A ~40-50 word summary that describes the context and significance of the findings for a general readership should be included on the title page. The statement should be written in the present tense and refer to the work in the third person. It should begin with "First author name(s) et al..." to match our preferred style.

10) Conflict of interest statement: JCB requires inclusion of a statement in the acknowledgements regarding competing financial interests. If no competing financial interests exist, please include the following statement: "The authors declare no competing financial interests." If competing interests are declared, please follow your statement of these competing interests with the following statement: "The authors declare no further competing financial interests."

11) A separate author contribution section is required following the Acknowledgments in all research manuscripts. All authors should be mentioned and designated by their first and middle initials and full surnames. We encourage use of the CRediT nomenclature (<https://casrai.org/credit/>).

12) ORCID IDs: ORCID IDs are unique identifiers allowing researchers to create a record of their various scholarly contributions in a single place. Please note that ORCID IDs are required for all authors. At resubmission of your final files, please be sure to provide your ORCID ID and those of all co-authors.

13) Journal of Cell Biology now requires a data availability statement for all research article submissions. These statements will be published in the article directly above the Acknowledgments. The statement should address all data underlying the research presented in the manuscript. Please visit the JCB instructions for authors for guidelines and examples of statements at (<https://rupress.org/jcb/pages/editorial-policies#data-availability-statement>).

B. FINAL FILES:

Thank you for your attention to these final processing requirements. Please revise and format the manuscript and upload materials within 7-10 days. If you need an extension for whatever reason, please let us know and we can work with you to

determine a suitable revision period.

Thank you for this interesting contribution, we look forward to publishing your paper in Journal of Cell Biology.

Sincerely,

Eva Nogales, PhD
Monitoring Editor
Journal of Cell Biology

Dan Simon, PhD
Scientific Editor
Journal of Cell Biology

Reviewer #1 (Comments to the Authors (Required)):

Authors have addressed most of my concerns. I still have several questions to be answered before formal acceptance.

1. To enhance signal-to-noise, the authors performed distance-weighted average of the signal of triangles with an adjustable radius (eg: 12 nm). Specifically, for the triangles within this adjustable radius of the edge of the edge-filtered surface, how to do the distance-weighted average?
2. Line 128-129: Base on the context of the paragraph, the authors may need to delete "We found that thickness variations".
3. Line 132: Change "9.98 Å/ Å/pixel" to "9.98 Å/pixel"
4. In Supplemental Figure 1C, segmentations generated from tomograms reconstructed at 13.3 Å/pixel are not present.

Reviewer #2 (Comments to the Authors (Required)):

The manuscript has been substantially improved. Regarding the differences between this work and the preprint by Glushkova et al., I agree that the results cannot be directly compared. Nevertheless, this also implies that the values measured by either method may not guarantee the 'absolute' membrane thicknesses measured, as there are intrinsic discrepancies between the two methods developed. The relative information reflection variations and distributions of membranes from different organelles and regions is still valuable.

Reviewer #1 (Comments to the Authors (Required)):

Authors have addressed most of my concerns. I still have several questions to be answered before formal acceptance.

1. To enhance signal-to-noise, the authors performed distance-weighted average of the signal of triangles with an adjustable radius (eg: 12 nm). Specifically, for the triangles within this adjustable radius of the edge of the edge-filtered surface, how to do the distance-weighted average?

To calculate the distance-weighted average: within the defined radius, each triangle's plot is first multiplied by $1/1+d$, where d is the distance from the center. The plots are then summed together. The full details of this procedure are provided in the Methods section on page 20, lines 20-26. We also present a graphical representation of the distance weighting in Figure 1D. We hope this clarification addresses the reviewer's question.

2. Line 128-129: Base on the context of the paragraph, the authors may need to delete "We found that thickness variations".

We thank the reviewer for finding this incomplete sentence. We have removed it in the updated version.

3. Line 132: Change "9.98 Å/ Å/pixel" to "9.98 Å/pixel"

We have corrected this in the updated version.

4. In Supplemental Figure 1C, segmentations generated from tomograms reconstructed at 13.3 Å/pixel are not present.

Thank you for bringing this to our attention. We have now added additional segmentations corresponding to tomograms reconstructed at 13.3 Å/pixel in Supplemental Figure 1C.

Reviewer #2 (Comments to the Authors (Required)):

The manuscript has been substantially improved. Regarding the differences between this work and the preprint by Glushkova et al., I agree that the results cannot be directly compared. Nevertheless, this also implies that the values measured by either method may not guarantee the 'absolute' membrane thicknesses measured, as there are intrinsic discrepancies between the two methods developed. The relative information reflection variations and distributions of membranes from different organelles and regions is still valuable.

We appreciate the reviewer's positive assessment and feel that the suggestions and feedback significantly strengthened the revised manuscript. We agree that caution is warranted when interpreting absolute membrane thickness values from either approach. However, we would like to emphasize that, in the revised manuscript, we benchmarked our method against an established 2D projection-based approach

using the same in vitro vesicle samples and obtained nearly identical results (Figure 5). We believe this comparison supports the accuracy of our three-dimensional membrane thickness measurements in cells.